# A linear pathway for inositol pyrophosphate metabolism revealed by $^{18}$O labeling and model reduction

Jacques Hermes[1,2,3]*, Geun-Don Kim[4], Guizhen Liu[3,5], Maria Giovanna De Leo[4], Andreas Mayer[4], Henning Jessen[3,5], Jens Timmer[1,2,3]

**1** Institute of Physics, University of Freiburg, Freiburg, Germany, **2** Freiburg Center for Data Analysis and Modelling (FDM), University of Freiburg, Freiburg, Germany, **3** Centre for Integrative Biological Signalling Studies (CIBSS), University of Freiburg, Freiburg, Germany, **4** Département d'immunobiologie, Université de Lausanne, Epalinges, Switzerland, **5** Institute of Organic Chemistry, University of Freiburg, Freiburg, Germany

* jacqueshermes42@gmail.com

## Abstract

The homeostasis of intracellular inorganic phosphate is essential for eukaryotic metabolism and is regulated by the INPHORS signalling pathway, which employs inositol pyrophosphates (IPPs) as key intermediary messengers. This study investigates the metabolic pathways of inositol pyrophosphates (IPPs) in the yeast cell line Pho$^{\Delta SPX}$ and the human tumor cell line HCT116. Utilizing pulse-labelling experiments with $^{18}$O water and ordinary differential equation (ODE) models, we explore the synthesis and turnover of the highly phosphorylated IPP, 1,5-InsP$_8$. Our findings challenge the notion that 1,5-InsP$_8$ can be synthesized through distinct routes, revealing a linear reaction sequence in both systems. Employing model reduction via the profile likelihood method, we achieved statistically concise identifiability analysis that led to significant biological insights. In yeast, we determined that 1,5-InsP$_8$ production primarily occurs through the phosphorylation of 5-InsP$_7$, with the pathway involving 1-InsP$_7$ deemed unnecessary as its removal did not compromise model accuracy. Crucially, this prediction of altered IPP concentrations was validated experimentally in *vip1Δ* and *kcs1Δ* knockout strains, providing orthogonal biological support for the reduced model. In HCT116 cells, 1,5-InsP$_8$ synthesis is mainly driven by 1-InsP$_7$, with variations observed across different experimental conditions. These results underscore the utility of model reduction in enhancing our understanding of metabolic pathways, coupling predictive modeling with experimental validation, and providing a framework for future investigations into the regulation and implications of linear IPP pathways in eukaryotic cells.

**Data availability statement:** All data and code used for running experiments, model fitting, and plotting is available on a GitHub repository at https://github.com/vandensich/The_break_in_the_cycle. We have also used Zenodo to assign a DOI to the repository: https://doi.org/10.5281/zenodo.15480075.

**Funding:** JT acknowledges funding by the Deutsche Forschungsgemeinschaft (DFG, German Research Foundation) – Project-ID 499552394 – CRC 1597 and by the state of Baden-Württemberg through bwHPC and the German Research Foundation (DFG) through grant INST 35/159. JT and HJ acknowledge funding by the Deutsche Forschungsgemeinschaft (DFG) under Germany's excellence strategy (CIBSS, EXC-2189, Project ID 390939984). H.J. acknowledges funding from the Volkswagen Foundation (VW Momentum Grant 98604). AM is grateful for support by the ERC (Project 788442) and the SNSF (project 10.001.312) and GK for support by HFSPO (LT000588/2019-L). The funders had no role in study design, data collection and analysis, decision to publish, or preparation of the manuscript. We acknowledge support by the Open Access Publication Fund of the University of Freiburg.

**Competing interests:** The authors have declared that no competing interests exist.

## Author summary

This research reveals how cells produce $1,5\text{-InsP}_8$, a key molecule in phosphate homeostasis, a vital part of cellular metabolism. Using isotope tracing and mathematical modelling, the study shows that both yeast and human cancer cells follow a linear, rather than branching, pathway to build this molecule. The analysis identified essential steps in the process, challenging previous assumptions about how these molecules are synthesized. By improving our understanding of inositol pyrophosphate metabolism, this work offers new insights into how cells regulate phosphate levels, with implications for cellular energy supply and metabolism in general.

## Introduction

All living organisms utilize signalling molecules to respond to external stimuli and maintain internal stability. Inositol pyrophosphates (IPPs) are a class of signalling molecules that is evolutionarily conserved in eukaryotic cells [1]. IPPs are generated as part of the INPHORS (intracellular phosphate reception and signalling) pathway (Ref [17]). They bind to a familiy of conserved SPX domains, change their conformation and/or association state, and thereby regulate a network of SPX-carrying target proteins [2–7]. These target proteins have in many cases a direct role in phosphate transport or utilisation. Thereby, INPHORS signalling is a key regulator of intracellular phosphate homeostasis in eukaryotic cells [8–10].

IPPs occurring in cells contain one or two pyrophosphate groups attached to the inositol ring. The number and positioning of the pyrophosphate groups may confer different physiological roles [11–13]. IPPs are synthesized by several kinases using $\text{InsP}_6$ as a backbone. Inositol hexakisphosphate kinases (IP6Ks in mammals; Kcs1 in yeast) add a phosphate group at the C5 position of the inositol ring, generating $5\text{-InsP}_7$ from $\text{InsP}_6$ or $1,5\text{-InsP}_8$ from $1\text{-InsP}_7$ [14,15]. Diphosphoinositol pentakisphosphate kinases (PPIP5Ks in mammals; Vip1 in yeast) phosphylate at the C1 position of the inositol ring and synthesize $1\text{-InsP}_7$ from $\text{InsP}_6$ or $1,5\text{-InsP}_8$ from $5\text{-InsP}_7$ [16,17]. Further IPPs have been identified in plants and mammalian cells [18,19], but at present it is not clear what physiological roles they might play. Highly phosphorylated IPPs are converted back to $\text{InsP}_6$ by several phosphatases. PPIP5Ks are bidirectional enzymes containing both kinase and phosphatase activities [20,21]. Their phosphatase domain can remove the $\beta$-phosphate at the C1 position from $1\text{-InsP}_7$ or $1,5\text{-InsP}_8$. In mammals, diphosphoinositol polyphosphate phosphohydrolases (DIPPs) nonspecifically remove the $\beta$-phosphate group, leading to the conversion of IPPs back to $\text{InsP}_6$ [22]. In yeast, several different phosphatases are involved in the dephosphorylation of IPPs. Ddp1, a yeast homolog of mammalian DIPPs, preferentially removes the $\beta$-phosphate group at the C1 position, generating $5\text{-InsP}_7$ from $1,5\text{-InsP}_8$ or $\text{InsP}_6$ from $1\text{-InsP}_7$ [23–25]. Meanwhile, Siw14 removes the $\beta$-phosphate at the C5 position to create $1\text{-InsP}_7$ from $1,5\text{-InsP}_8$ or $\text{InsP}_6$ from $5\text{-InsP}_7$ [26–28].

The levels of IPPs change quantitatively and qualitatively in response to various cellular signals and environmental stimuli. For example, when yeast cells are starved for inorganic phosphate ($P_i$), the amount of overall IPPs decrease rapidly, with 1,5-InsP$_8$ showing a faster and more pronounced decline than 1-InsP$_7$ and 5-InsP$_7$ [29]. The decrease in 1,5-InsP$_8$ triggers the phosphate-responsive signal transduction (PHO) pathway. This pathway induces transcription of genes that increase the capacity for scavenging $P_i$ from the surrounding environment, and for liberating $P_i$ from polyphosphates, nucleic acids, or lipids to enable internal $P_i$ recycling [10]. 1,5-InsP$_8$ responds similarly to $P_i$ starvation in mammalian cells and in plants, suggesting that its role in $P_i$ signalling is conserved [30–32]. The amount of IPPs also changes during the cell cycle, perhaps as a consequence of fluctuating $P_i$ utilization [33–36]. Changes of IPPs in turn affect the cell cycle progression, e.g., by controlling kinetochore formation [37]. In addition, environmental stresses such as hyperosmotic stress [38], oxidative stress [39,40], and heat stress [41] can affect IPPs levels.

1,5-InsP$_8$ can be produced through two separate routes from InsP$_6$, via 5-InsP$_7$ or 1-InsP$_7$. It was proposed that the main route for synthesizing 1,5-InsP$_8$ from InsP$_6$ is through 5-InsP$_7$, while the conversion of InsP$_6$ to 1-InsP$_7$ by PPIP5Ks is not efficient *in vivo* [42]. This model was derived from several lines of evidence: 1) The cellular concentration of InsP$_6$ is markedly higher than that of other IPPs, but its turnover rate is significantly slower compared to that of other IPPs [43], implying that a substantial amount of InsP$_6$ is compartmentalized or remains unreactive. 2) Several studies showed that PPIP5Ks prefer 5-InsP$_7$ as a substrate over InsP$_6$ [44,45]. 3) The concentration of 1-InsP$_7$ within the cell is very low, almost to the point of being undetectable under normal conditions [29,46]. While substrate preferences of the IPP enzymes could be dissected through *in vitro* studies [44,45], such information does not automatically predict the metabolite fluxes *in vivo*. Therefore, it is required to analyse IPPs synthesis *in vivo* to understand the 1,5-InsP$_8$ synthesis pathway. Here we used data from pulse-labelling experiments with $^{18}$O water to dissect the kinetics of IPP turnover and ordinary differential equation (ODE) models to reveal the relevant metabolite fluxes. In order to estimate the parameters of our ODE models, we used the modelling tool box *dMod* [47], which was developed in our group, to fit the ODE model to the pulse-labelling data. Furthermore we applied a method called profile likelihood [48,49] to perform a statistical analysis of the estimated parameters' uncertainties. Based on this we reparameterized our models, i.e. adjusted the model structures, in a process called model reduction [50,51]. Our work was designed to test wether both the 1-InsP$_7$ and 5-InsP$_7$ branches are required to reproduce the labeling dynamics. If one branch can be consistently eliminated during model reduction without loss of fit, then a linear topology provides the most parsimonious explanation consistent with the data. Using this approach, we found that the analysed organisms, the yeast *Saccharomyces cerevisiae* and a human tumor cell-line (HCT116), synthesize 1,5-InsP$_8$ through distinct routes. In both cases however, the statistically preferred version of their IPP metabolic routes is not a cycle but rather a linear reaction network.

## Results

In this work we analysed two different biological organisms, baker's yeast and HCT116. The data sets were first analysed using the same initial, in the following called "full", mathematical model (Fig 1). The data-sets analysed in this manuscript stem from a recently published approach, in which cellular IPs are rapidly pulse-labelled through $^{18}$O water [52,53]. All experiments contain three biological replicates per time point. Since it is expected that the metabolic cycle has different kinetics between both organisms, the model was fitted and reduced for both organisms independently. However, the applied methods and strategies were identical in all analyses to ensure statistically sound results. For every iteration a multi-start fit approach was chosen, i.e. multiple fits with individual and randomly sampled initial parameter vectors were performed. To obtain statistically sound and comparable results, 1000 multi-starts were performed for each iteration. A schematic overview of the reduction procedure is given in Fig 2. The process begins by fitting the full model to the experimental data (A). Profile likelihood analysis is then performed to assess parameter identifiability (B). Non-identifiable parameters are reduced either by structurally modifying the underlying rate equation or by removing the corresponding reactions (C). The resulting reduced model is subsequently refitted, and the procedure is iterated until all parameters are

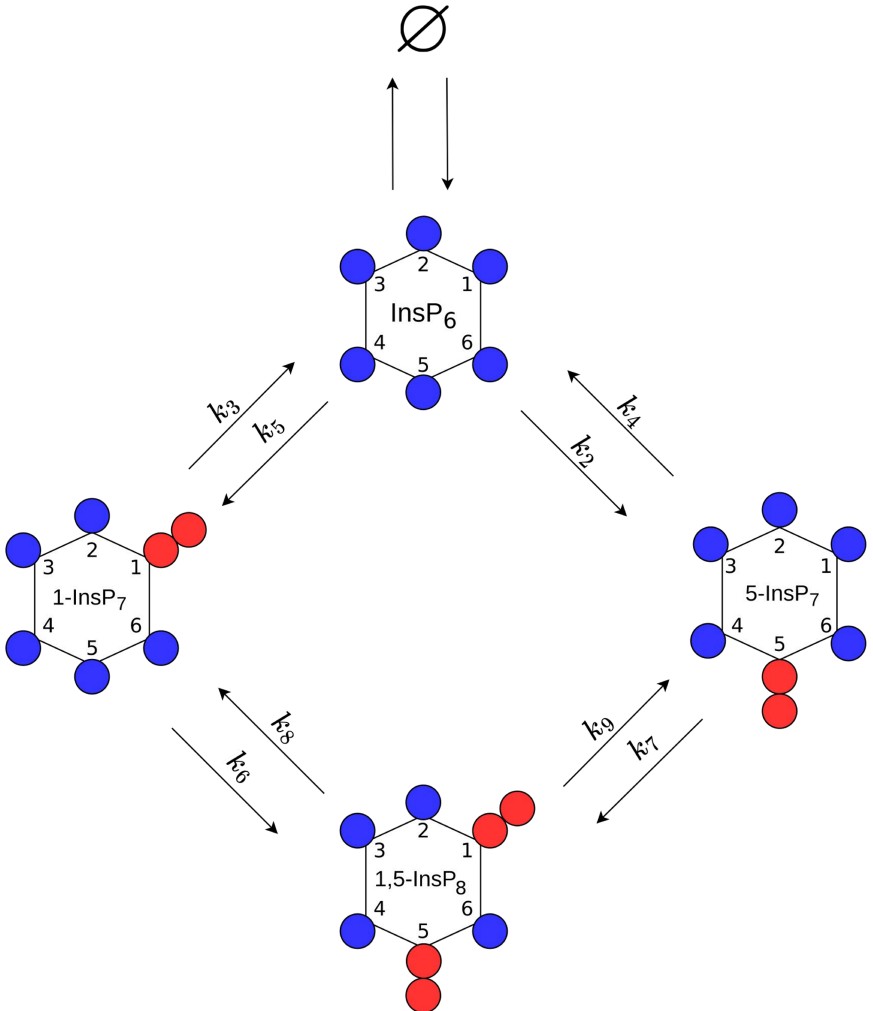

**Fig 1. Scheme of the cyclic production pathway of 1,5-InsP$_8$ upon which the "full" ODE-model is based.**

identifiable and a parsimonious model structure is obtained (D). In the following, the model reduction steps leading to the final fully reduced model are described for both organisms independently. The mathematical modelling techniques employed in this work are explained in detail in the Materials and Methods section.

## Yeast

In the yeast dataset, the levels of 1-InsP$_7$ were not shown because of their low abundancein the samples [52]. For that reason, we reinvestigated the $^{18}$O labelling of IPPs under the high P$_i$ conditions in the cytosol, which favours the accumulation of all IPP species [29]. To this end, we overexpressed the truncated form of the low-affinity phosphate transporter Pho90 (Pho90$^{\Delta SPX}$), which lacks the restraining SPX domain and allows enhanced P$_i$ import into the cells [54]. The cells were first grown in low phosphate medium (0.2 mM) and then transferred to high phosphate medium (50 mM) with 50% $^{18}$O water to generate P$_i$ influx and enhanced IPP synthesis. For further details on the generation of the data set, we refer to the Materials and Methods section. In a first step, the yeast data-set was analysed using the "full" ODE model, which yielded a set of best-fit parameters which adequately described the dynamics observed in the data and yielded a

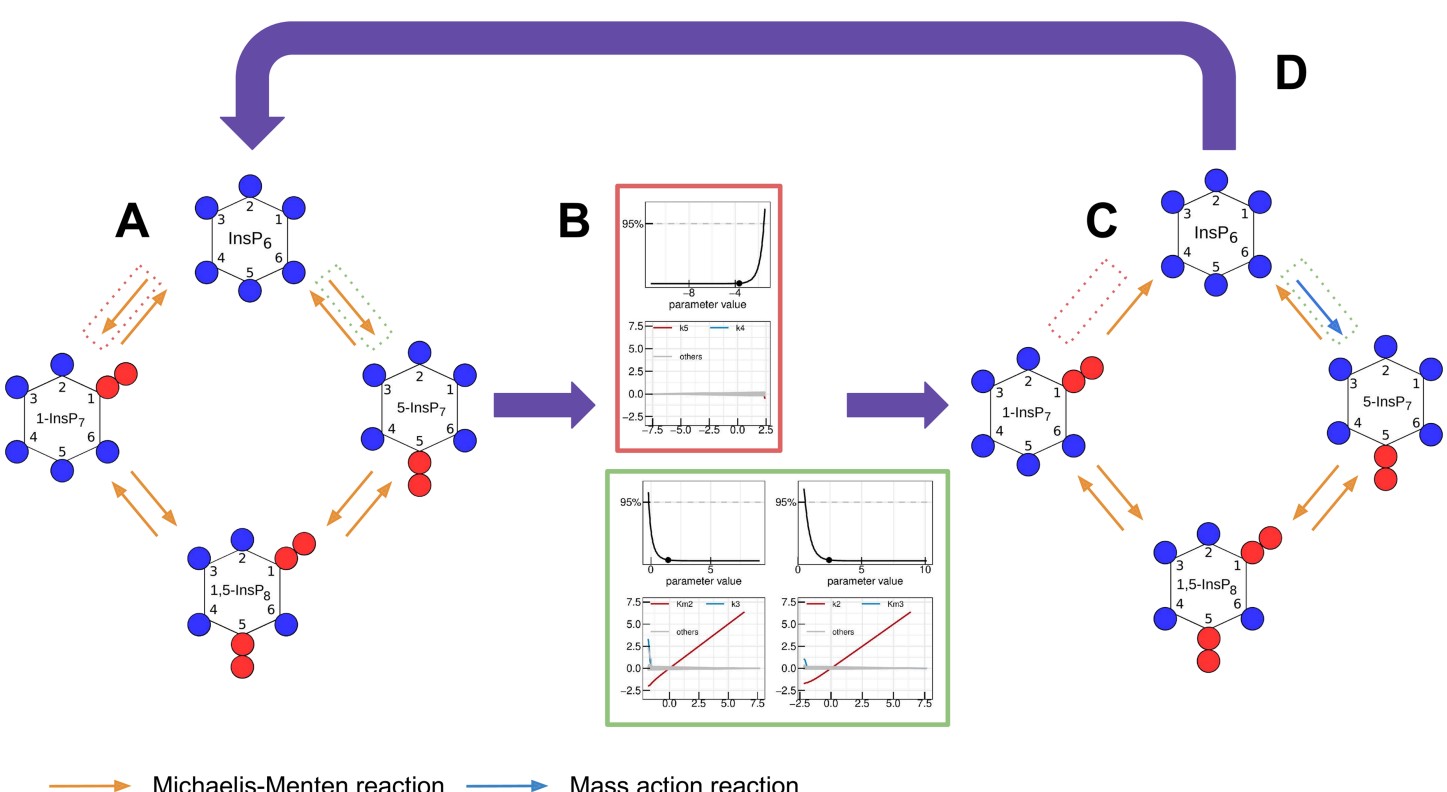

**Fig 2**. **Schematic overview of the model reduction procedure.** (A) Full model fitted to the experimental data. All reactions are initially represented as Michaelis-Menten rate laws (orange arrows, see legend at bottom). Two reactions are highlighted: one enclosed in a red dotted box, one in a green dotted box. (B) Profile likelihoods calculated to assess parameter identifiability. The reaction in the red box is identified as non-contributing, while the parameters in the green box are identified as coupled. (C) Reduced model after this step. The reaction in the red box has been removed, while the reaction in the green box has been simplified to a mass-action process (blue arrow). (D) Iterative procedure. The reduced model is refitted, and steps (A–C) are repeated until all parameters are identifiable and a parsimonious model structure is obtained.

Bayesian information criterion value (BIC) of -1011,9. The data as well as the time-courses obtained via the mathematical model with best-fit parameters are shown in panel A of Fig 3. In panel B, the likelihood values of the 200 best fits out of the 1000 multi-starts are shown sorted by the lowest likelihood. Such a plot, called waterfall plot, indicates that the best fit found in the analysis is found by eight different initial parameter configurations, providing evidence that the found optimum is in fact a global and not just a local one [48]. In a next step, parameter profiles were calculated, as shown in Fig 3C. This resulted in six model reactions where the rate parameters $k_i$ and the Michaelis-Menten (MM) constant $K_{M_i}$ are both not identifiable, namely $i \in \{2, 3, 5, 6, 7, 8\}$. A deviation from the profiled parameters best-fit value can thus be compensated by the re-optimization of other parameter values resulting in the same likelihood values, therefore being called coupled to or dependent on the compensating parameters. The dependencies of the non-identifiable $k_i$s and $K_{M_i}$s are graphically displayed in Fig 4 where the two strongest dependencies are displayed as paths in red and blue respectively and the dependencies of the remaining model parameters are shown in gray. In most panels, only the strongest dependency in red is visible since that parameter only strongly depends on one other parameter and the gray lines are thus overlapping the blue line. The parameter profile of interest is again plotted above its respective paths. Except for $k_6$ and $k_8$, the parameters solely depend on the $K_M$ or $k$ values of their respective reactions. Furthermore, if the profiled parameter value is increased the respective counterpart also increases in order to compensate for the change. Combining this observation with the fact that the $K_M$ values of interest have profiles with confidence intervals open to $+\infty$ we see they can

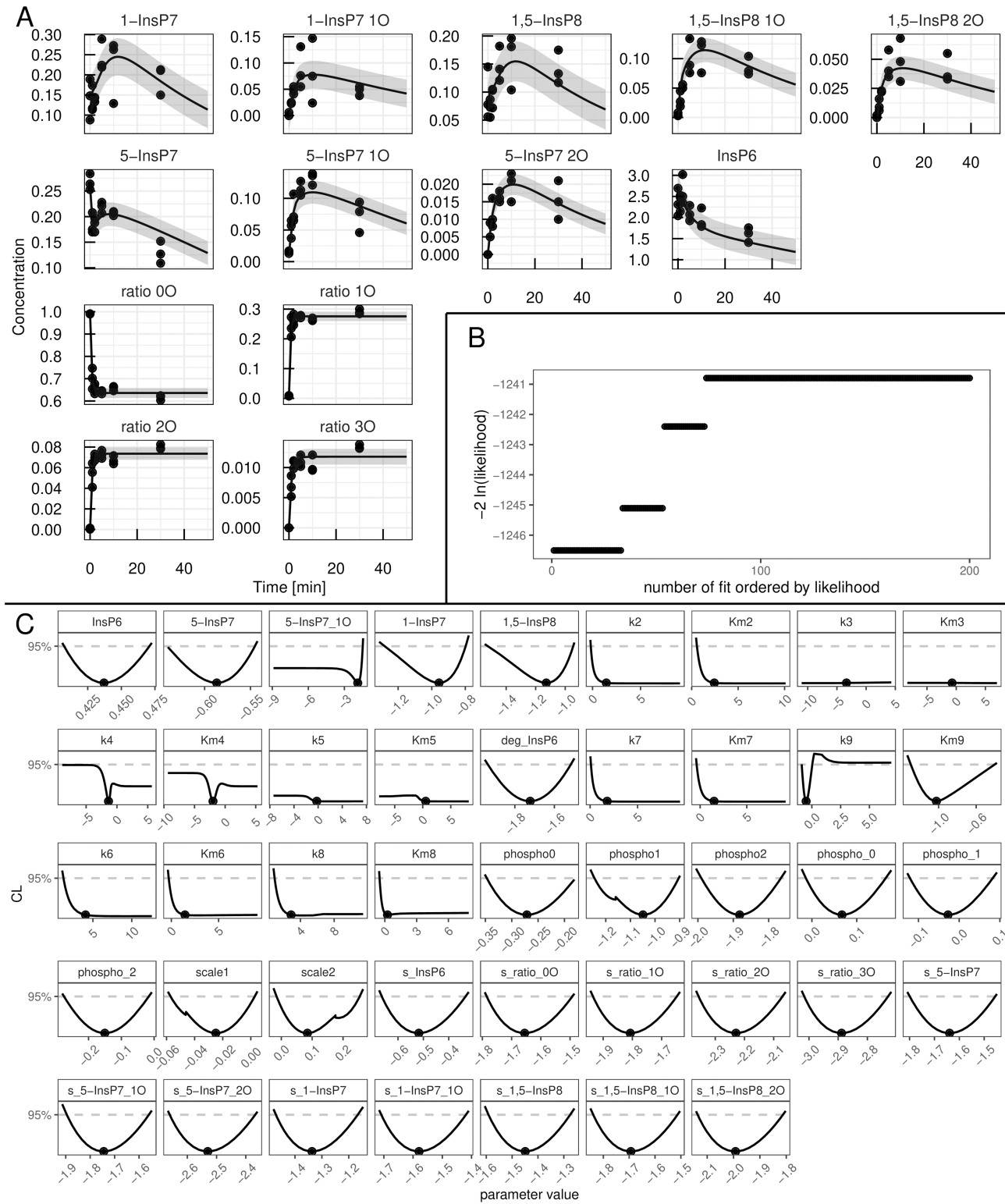

**Fig 3. Analysis of the yeast data-set with the "full" mathematical model.** (A) Fit of the mathematical model with the best-fit parameters (line) to the experimental data (points). The values are given in $\mu$M except for the four species in the lower left of the panel which represent ratios of 1,2,3-labelled to 0-unlabelled ATP $\gamma$-phosphate. (B) Likelihood values of the 200 best multi-start fits, ordered by lowest likelihood value. (C) Profiles of the best-fit parameters of the full model (line). Best-fit parameter value are represented as points. Parameter values (x-axis) are displayed on log10 scale.

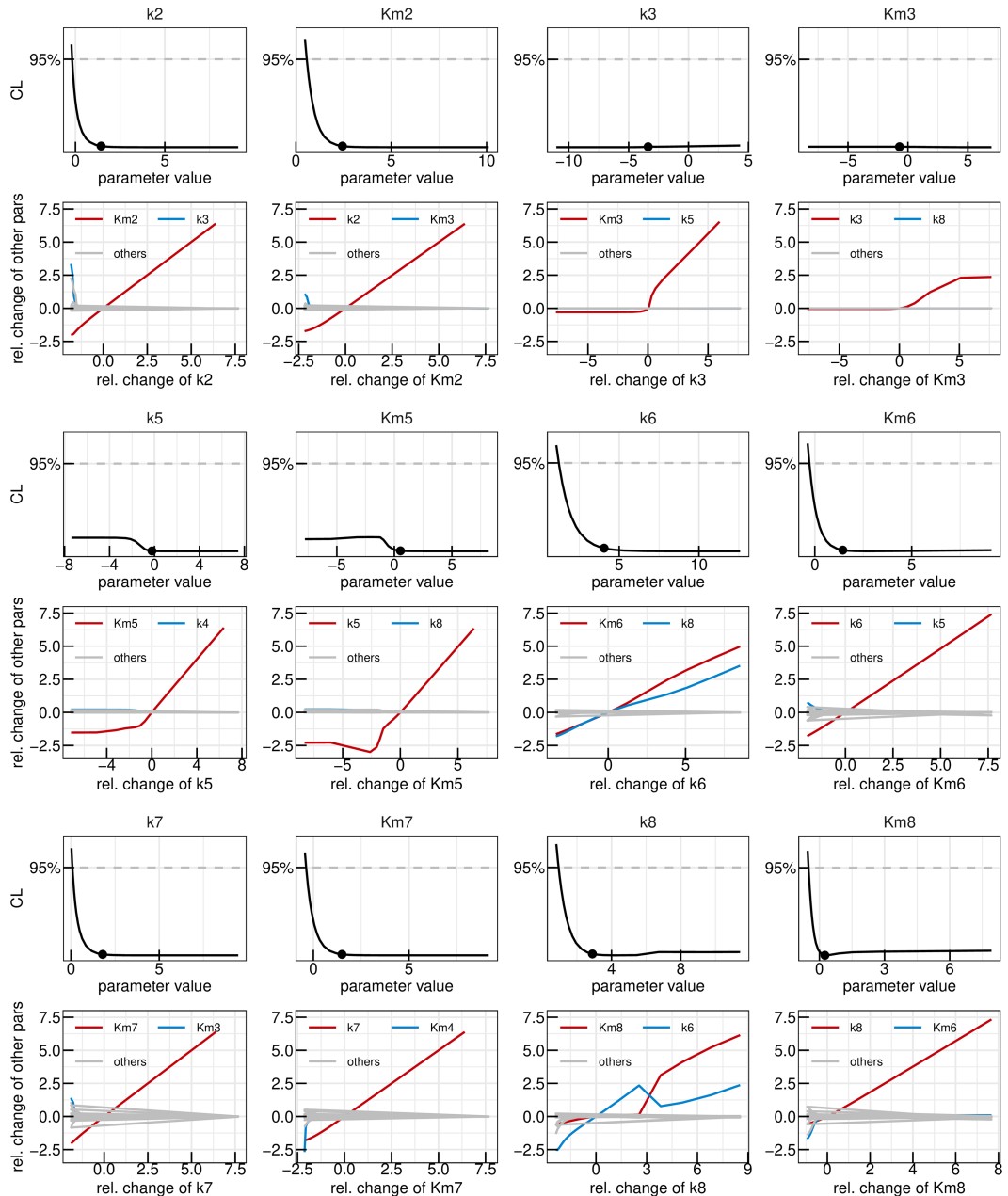

**Fig 4**. **First reduction step in the yeast model.** Profiles of the parameters considered for the first model reduction step in the yeast metabolic cycle. Best-fit parameter value are represented as points. Parameter values (x-axis) are displayed on log10 scale. Below each profile, the dependencies of all other parameters on the profiled parameter are plotted. Coupling strength is quantified by the relative change of a secondary parameter from its best-fit value after re-optimizing all parameters at each fixed value of the profiled parameter (x-axis). Both axes are shown on a $\log_{10}$ scale. The strongest dependency is highlighted in red, the next strongest in blue, and the remaining dependencies in gray.

take very large values without worsening the likelihood. This supports reducing these reactions to mass-action reactions. Such mass-action reactions occur in a regime of Michaelis-Menten (MM) reactions where the $K_M$ value is much larger than the available substrate concentration [55]. For $k_6$ and $k_8$ this reduction is also valid, since their $K_M$ values, which are

the parameters that are being reduced, only depend on their respective $k$ values. However, we do not expect $k_6$ and $k_8$ to become identifiable via this step, due to their strong dependencies upon each other, as seen in Fig 4.

A subsequent multi-start fit of the so reduced model to the same data set showed no visible changes in goodness-of-fit, while BIC improved to -1045,2. Examining the parameter profiles resulting from this refitting implied that also the reaction rates $k_3$ and $k_5$ could be set to zero, since their profiles are open towards $-\infty$ on the log scale (or zero on the linear scale), as shown in Fig 5A. Furthermore, no significant dependencies for small values of these parameters are present, demonstrating that for values smaller then the best-fit one, other parameters do not need to be adjusted in order to keep the best likelihood value, justifying setting these reaction rates to zero on linear scale. The profiles of both $k_8$ and $k_6$ on the other hand are still open towards $+\infty$ on the log scale, meaning both reactions could be set to arbitrarily high values without significantly changing the model's capacity of describing the data. The remaining dependencies of the two parameters in Fig 5A indicate that they linearly dependent upon one-another signifying that one can introduce the parameter transformation $k_6 = \frac{k_6}{k_8} \cdot k_8 = \text{ratio}_{k_6,k_8} \cdot k_8$. It is expected, that the ratio between the two rates becomes identifiable.

The now twice reduced model was once again fitted to the data, yielding an unchanged description of the data while resulting in a BIC of -1055,9. In a last reduction step, the reaction involving $k_4$ and $K_{M_4}$ are reduced to mass action, following the same reasoning as for the other mass-action reduction steps. Additionally, the initial value for single labelled 5-InsP$_7$ was also reduced to 0 following the reasoning of the previous reductions of $k_3$ and $k_5$. The profiles and dependencies of these three parameters are shown in Fig 5B, together with the identifiable profile of the ratio between $k_6$ and $k_8$ and the profile of $k_8$ which, as expected, is still open towards $+\infty$. Since the goodness-of-fit is not limited by large values of $k_8$ its value was fixed to $10^5$ on the linear scale. The final, completely reduced model yielded profiles shown in Fig 6A and a BIC of -1069.5. It is completely identifiable as seen in the parabolic shapes of every single calculated profile. Note, that only the description of the reaction involving $k_9$ requires the use of Michaelis-Menten kinetic since both $k_9$ and $K_{M_9}$ are identifiable parameters. In panel B of Fig 6 the waterfall plot of the 200 best multi-starts is plotted, where no step can be observed since all 200 fits ended up at the same best likelihood value, which indicates high trust in the employed numerical optimisations. To further strengthen the confidence in the obtained optimum, the distribution for the parameters of the 200 best runs is displayed in S7 Fig which shows a point like distribution for the parameters across the selected fits. Furthermore a time vs residual plot is shown in Fig S6 Fig providing no evidence of systematic bias of the residuals over time thus indicating that the fits are not unduly driven by individual noisy points.

Finally, in panel C, a scheme of the statistically favoured version of the reaction pathway is depicted, with red crosses marking the reactions which were reduced to 0. This scheme is not a cycle anymore, but rather a chain where production of 1-InsP$_7$ can only be facilitated via dephosphorylation of 1,5-InsP$_8$ rather than phosphorylation of InsP$_6$.

## Knockout validation of the reduced yeast model

To move beyond statistical support and test whether our reduced yeast topology is a valid biological prediction, we examined IPP concentrations in wild type, *vip1Δ*, and *kcs1Δ* strains under the same high phosphate conditions used for isotope tracing.

The reduced model predicts that the InsP$_6 \rightarrow$ 1-InsP$_7$ branch is dispensable, implying that 1-InsP$_7$ should not accumulate if Kcs1 is absent. In contrast, the canonical cyclic model predicts that Vip1 can still generate 1-InsP$_7$ independently of Kcs1. Thus, the two topologies make distinct, testable predictions for the *kcs1Δ* strain.

As summarized in Fig 7, Kcs1 catalyzes the InsP$_6 \rightarrow$ 5-InsP$_7$ and 1-InsP$_7 \rightarrow$ 1,5-InsP$_8$ reactions, whereas Vip1 catalyzes InsP$_6 \rightarrow$ 1-InsP$_7$ and 5-InsP$_7 \rightarrow$ 1,5-InsP$_8$. Consistent with these assignments, the *vip1Δ* strain displayed the expected elevation of 5-InsP$_7$ and reduction of 1,5-InsP$_8$. However, the *kcs1Δ* strain exhibited no detectable 1-InsP$_7$.

This outcome is difficult to reconcile with the cyclic model, which would predict 1-InsP$_7$ production via Vip1, but it is fully consistent with the reduced model in which the InsP$_6 \rightarrow$ 1-InsP$_7$ route is removed. The knockout data therefore provide

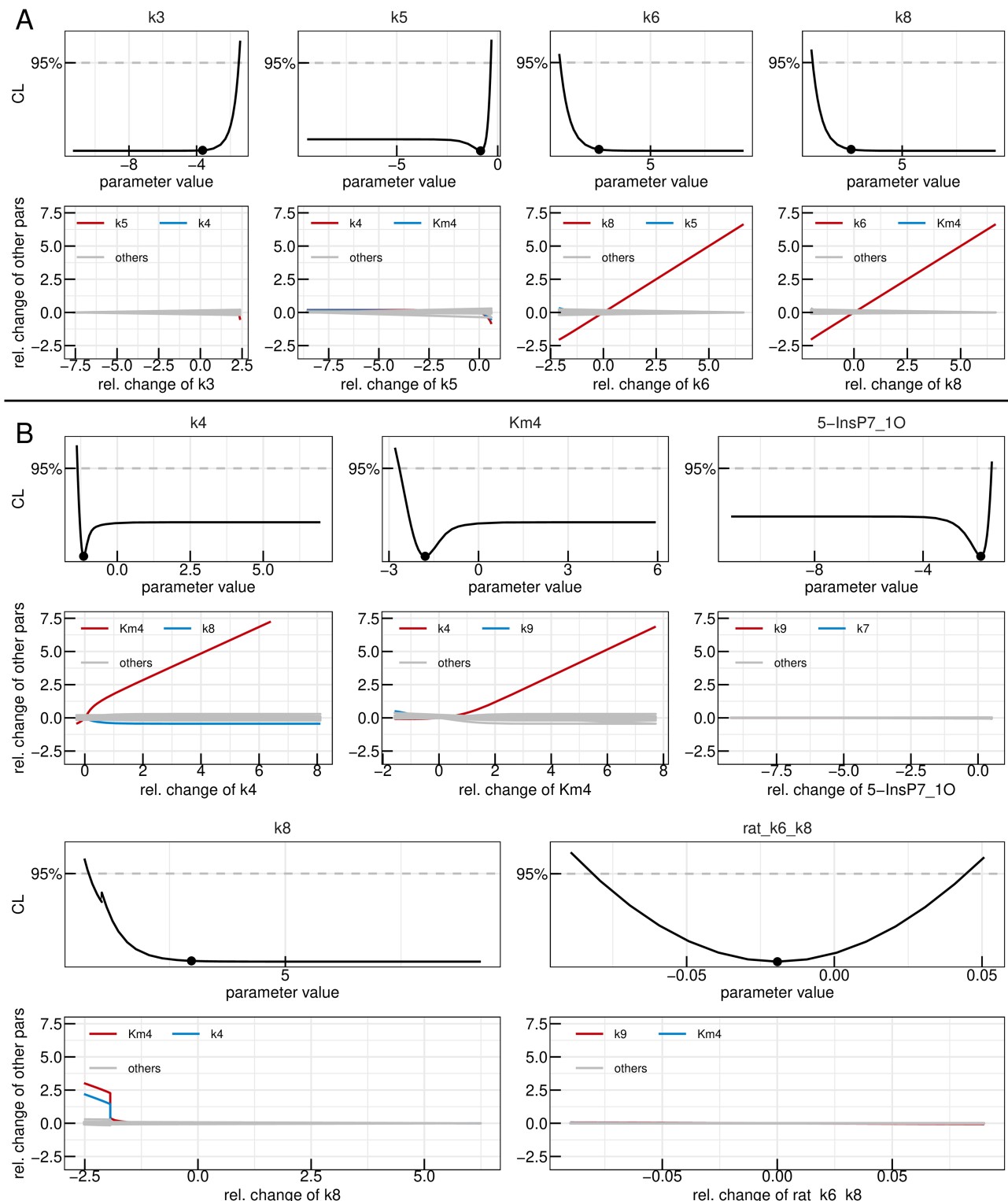

**Fig 5**. **Second and third reduction steps in the yeast model.** Profiles of the parameters considered for the second (A) and third (B) model reduction step in the yeast metabolic cycle. Best-fit parameter value are represented as points. Parameter values (x-axis) are displayed on log10 scale. Below each profile, the dependencies of all other parameters on the profiled parameter are plotted. Coupling strength is quantified by the relative change of a secondary parameter from its best-fit value after re-optimizing all parameters at each fixed value of the profiled parameter (x-axis). Both axes are shown on a $\log_{10}$ scale. The strongest dependency is highlighted in red, the next strongest in blue, and the remaining dependencies in gray.

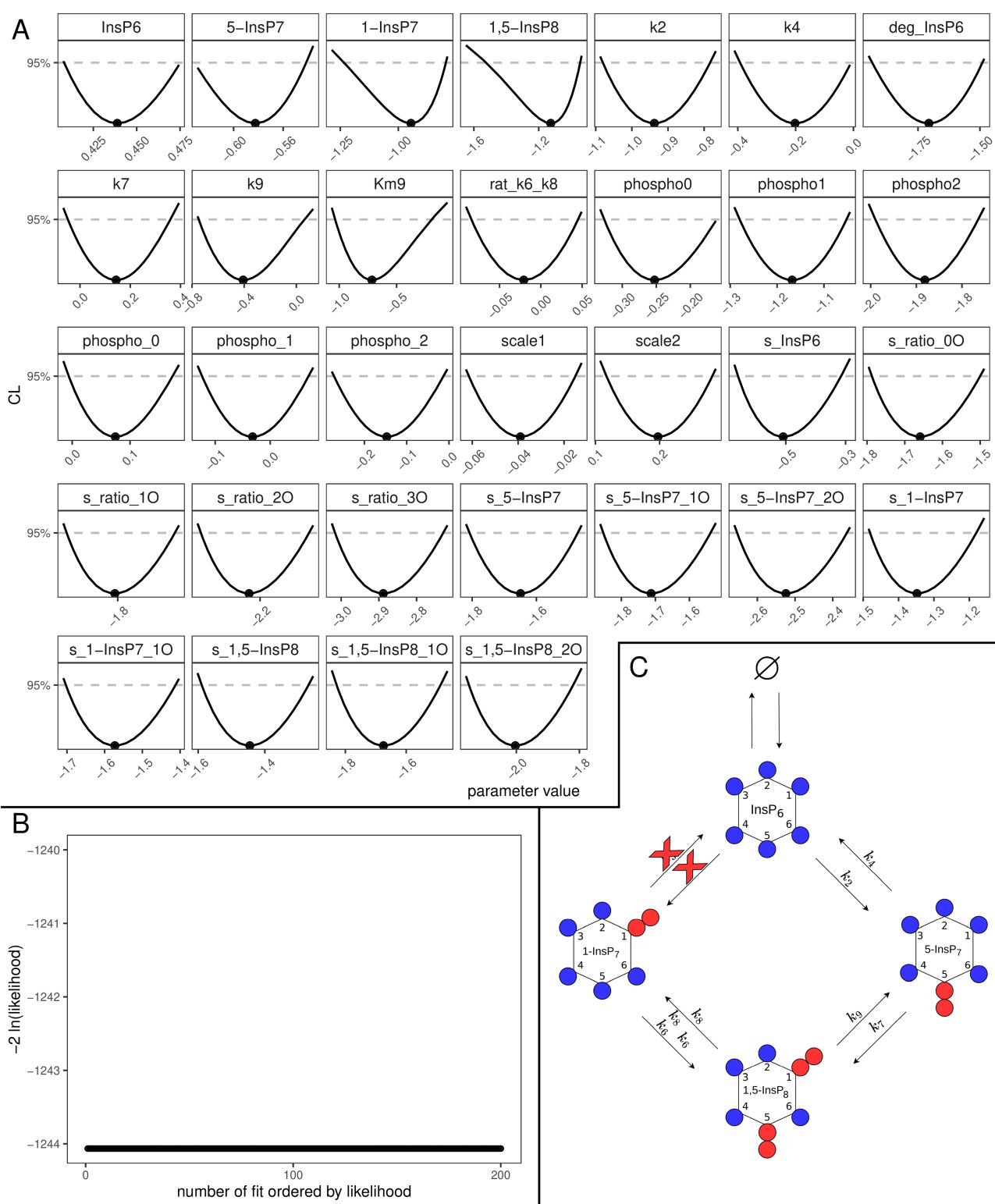

**Fig 6. Fully reduced yeast model.** (A) Profiles of the best-fit parameters of the reduced model (line). Best-fit parameter value are represented as points. Parameter values (x-axis) are displayed on log10 scale. (B) Likelihood values of the 200 best multi-start fits, ordered by lowest likelihood value. (C) Representation of the statistically favored transition scheme, displaying a chain-like pattern rather than a cycle.

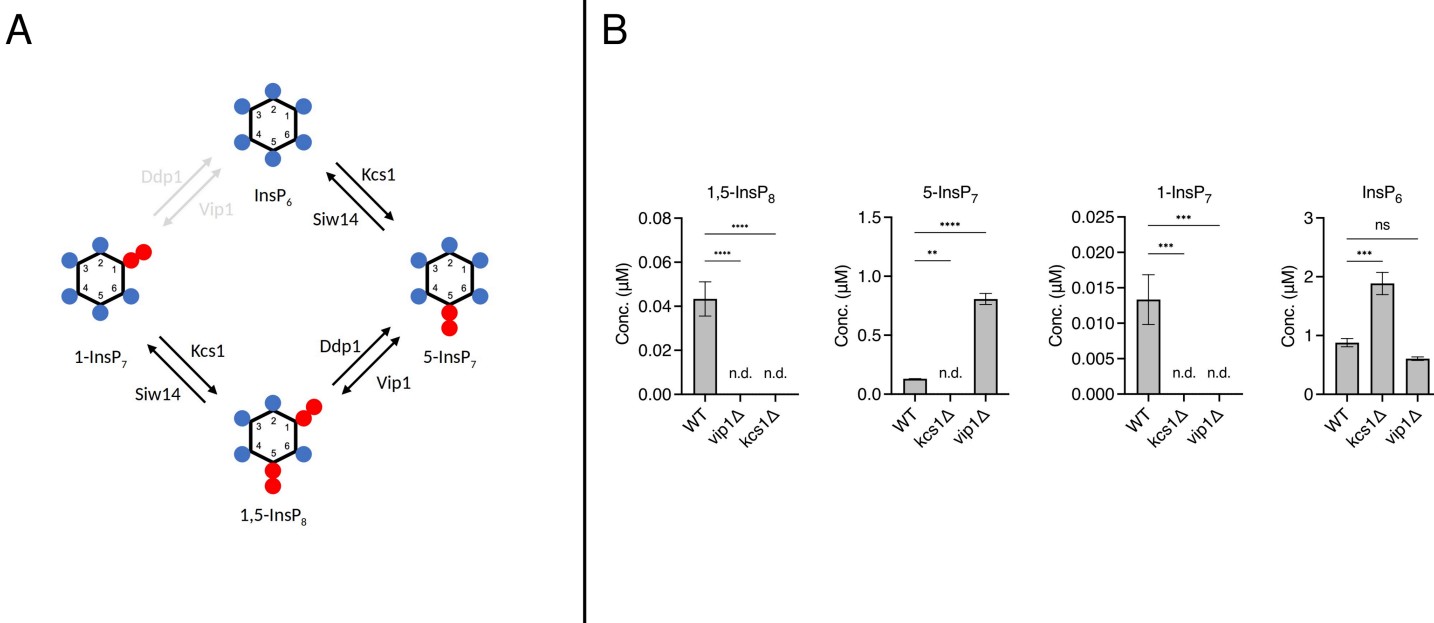

**Fig 7**. **Knockout validation of the reduced yeast model.** A) Schematic of the canonical cyclic pathway with enzymatic contributions of Kcs1 and Vip1. The Vip1-mediated $InsP_6 \rightarrow 1\text{-}InsP_7$ link is shown in grey, indicating that it is eliminated by model reduction and unsupported by knockout data. B) Concentrations of IPP species in wild type, $vip1\Delta$, and $kcs1\Delta$ strains under high phosphate conditions. Values are given in $\mu$M (y-axis) for each organism type (x-axis: WT, $vip1\Delta$, $kcs1\Delta$). Bars show mean values with error bars ($\pm$ SD, $n = 3$ biological replicates). Statistical analysis was performed using Tukey's multiple comparison test; ****, $p < 0.0001$; ***, $p < 0.001$; **, $p < 0.01$; *, $p < 0.05$, ns ; not significant. n.d. not detected.

independent biological evidence for the same break in the cycle predicted by the model reduction, validating the conclusion that under the tested conditions yeast IPP metabolism operates as a linear rather than cyclic pathway.

## HCT116

Human HCT116 cells were grown in phosphate-replete phosphate medium and then transferred to the same phosphate medium with 50% $^{18}O$ water. Further details on the generation of the data sets are given in Materials and Methods. The first fit with the "full" model yielded again an adequate description of the dynamics present in the data with a BIC of 164,6. The HCT116 data set together with the "full" model fit is depicted in Fig 8A. A waterfall plot displaying the sorted likelihood values of the 200 best fits is shown in Fig 8B, where eleven different initial configurations ended up in the same global optimum. Analysis of the model parameters profiles, displayed in Fig 8C, shows that also in this case a number of parameters are unidentifiable, indicated by a non-parabolic profile. Thus model reduction is required. The parameters of interest in the first model reduction step together with their respective paths can be found in Fig 9. The reactions involving $k_2$, $k_4$ and $k_5$ are reduced to mass action following the reasoning established in section Yeast. Furthermore the parameters $k_7$ and $K_{M_3}$ are set to zero, eliminating the transition involving $k_7$ from the model, while reducing the $k_3$ reaction to a zero order reaction. In a zero order reaction, as explained in section Mathematical modelling, the reaction rate is independent of the substrate concentration.

Refitting the HCT116 data-set with the reduced model yielded no visible changes in data description but resulted in an improved BIC of 128,9. In Fig 10A, the reduction to mass-action reactions can be appreciated for the $k_8$ and $k_9$ transitions. Even though, $K_{M_8}$ shows a strong dependency on $k_9$ for small values, for large values $K_{M_8}$ only depends linearly on its corresponding reaction rate, legitimising this reduction. Similar to $k_6$ and $k_8$ in section Yeast, here $k_3$ and $k_5$

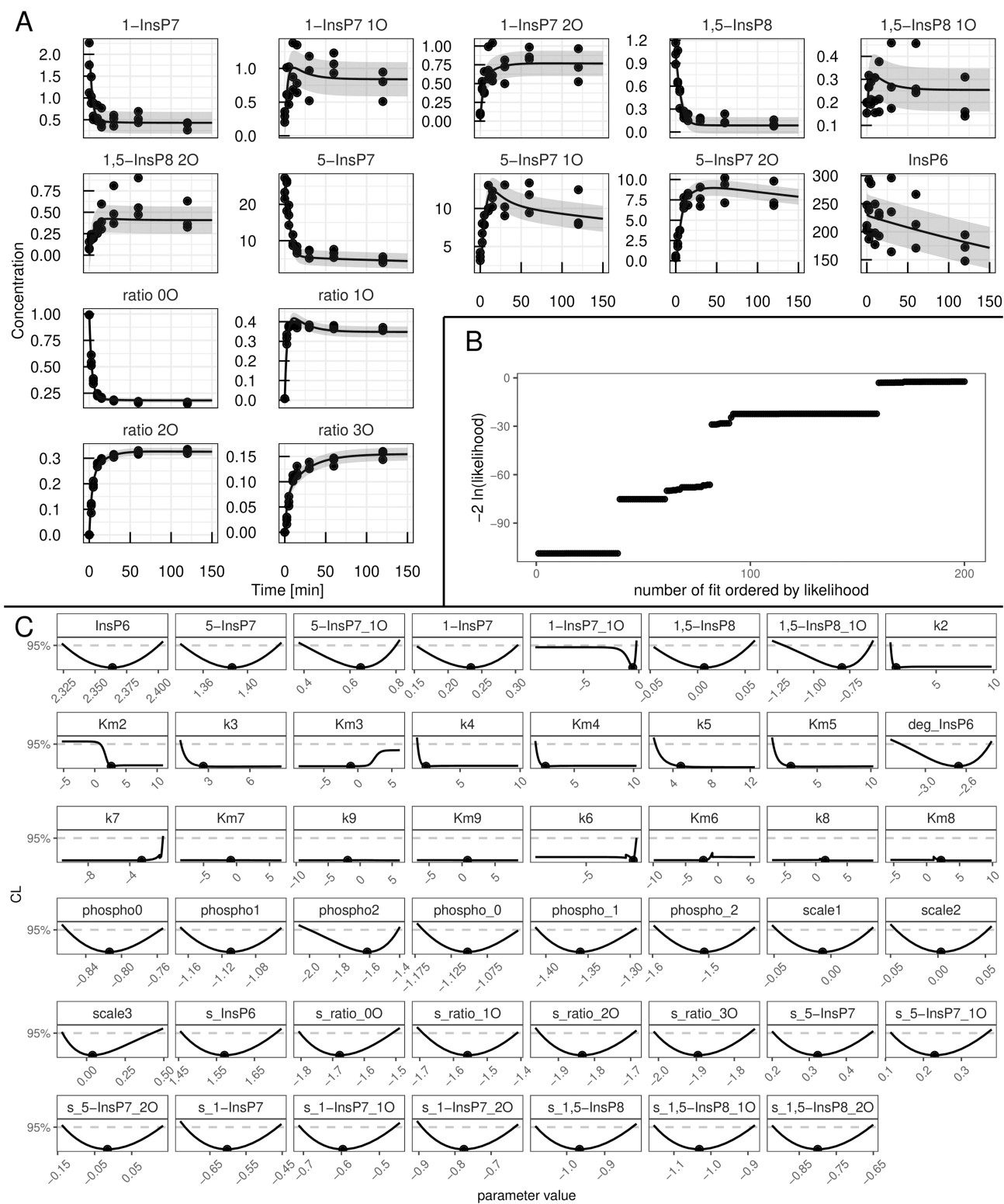

**Fig 8. Analysis of the HCT116 data-set with the "full" mathematical model model.** (A) Fit of the mathematical model with the best-fit parameters (line) to the experimental data (points). The values are given in $\mu$M except for the four species in the lower left of the panel which represent ratios of 1,2,3-labelled to 0-unlabelled $\gamma$-phosphate. (B) Likelihood values of the 200 best multi-start fits, ordered by lowest likelihood value. (C) Profiles of the best-fit parameters of the full model (line). Best-fit parameter value are represented as points. Parameter values (x-axis) are displayed on log10 scale.

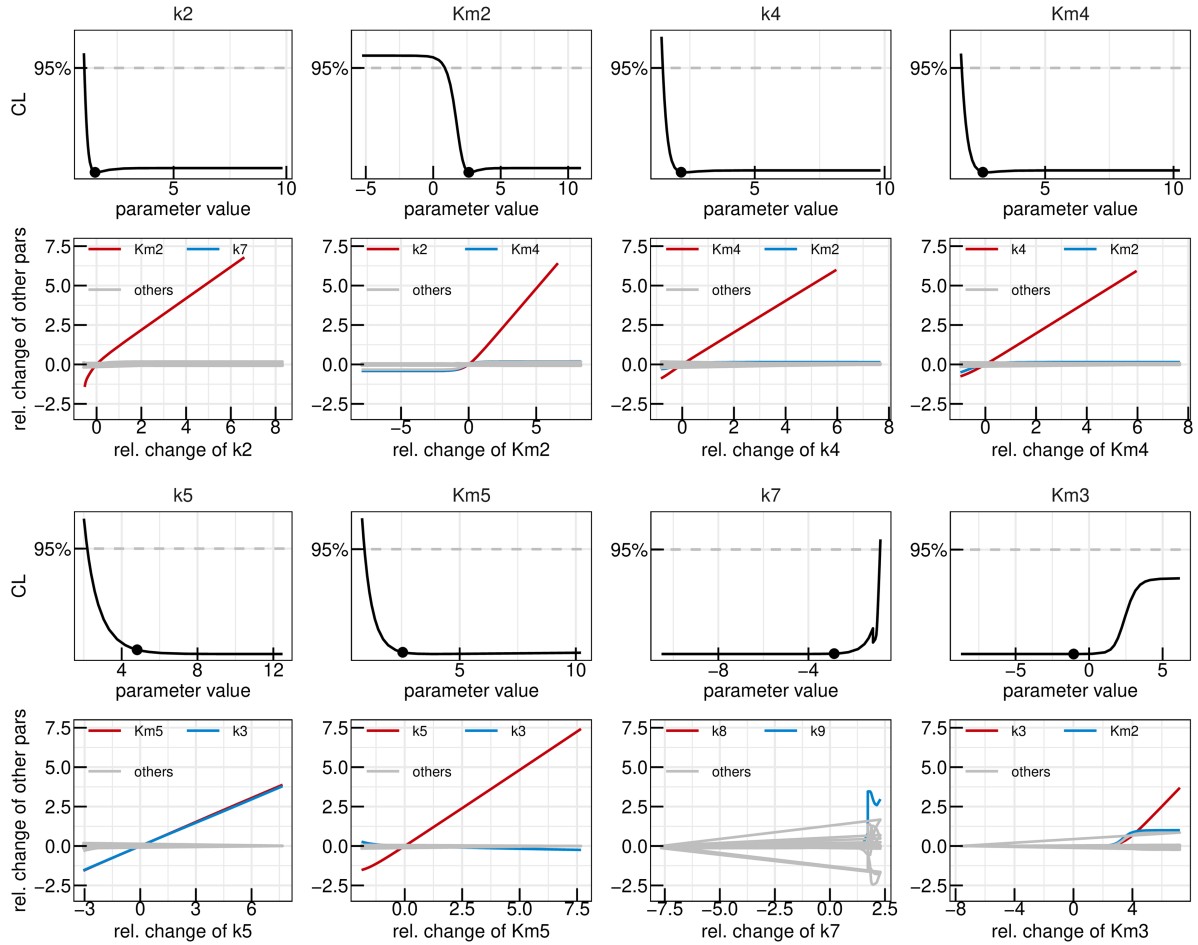

**Fig 9. First reduction step in the HCT116 model.** Profiles of the parameters considered for the first model reduction step in the yeast metabolic cycle. Best-fit parameter value are represented as points. Parameter values (x-axis) are displayed on log10 scale. Below each profile, the dependencies of all other parameters on the profiled parameter are plotted. Coupling strength is quantified by the relative change of a secondary parameter from its best-fit value after re-optimizing all parameters at each fixed value of the profiled parameter (x-axis). Both axes are shown on a $\log_{10}$ scale. The strongest dependency is highlighted in red, the next strongest in blue, and the remaining dependencies in gray.

depend linearly on each other while having profiles which are open towards $+\infty$, demanding the parameter transformation $k_5 = \frac{k_5}{k_3} \cdot k_3 = \text{ratio}_{k_5,k_3} \cdot k_3$. Lastly, the reaction involving $k_6$ is reduced to a zero order reaction following the example of $k_3$.

This version of the model can describe the data without a visible change in goodness-of-fit, but yielded a BIC of 111,1. In a last reduction step, the parameters shown in Fig 10B are considered. $k_3$ is set to $10^5$, while the initial value of single labelled 1-InsP$_7$ is set to 0. When choosing between $k_8$ and $k_9$, the latter is set to zero, since it shows negligible dependence on $k_8$ compared to vice versa.

The final fit of the fully reduced model to the data generated a BIC of 96,94. The fully identifiable model parameters are shown in Fig 11A. In this panel, the identifiable ratio between $k_5$ and $k_3$ can be appreciated as well as the fact that all of the remaining reactions could be reduced to mass action while still describing the data accurately. In the final waterfall plot of the fully reduced model (Fig 11B 26 fits ended up in the global optimum. Thus also the IPP metabolic cycle of the HCT116 appears to be linear rather than cyclic (Fig 11C), where 1,5-InsP$_8$ is solely generated by phosphorylation of 1-InsP$_7$ .

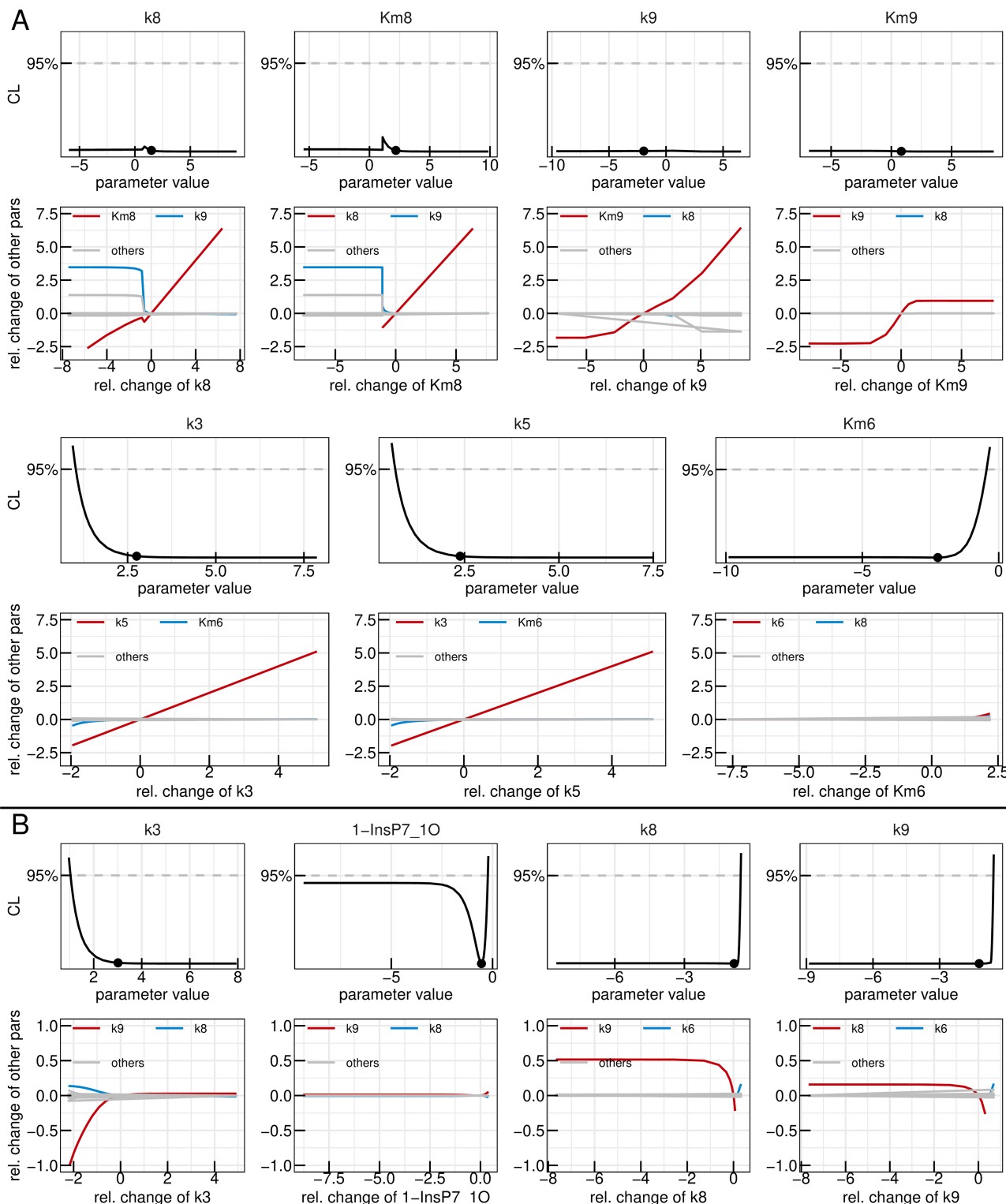

**Fig 10. Second and third reduction steps in the HCT116 model.** Profiles of the parameters considered for the second (A) and third (B) model reduction step in the yeast metabolic cycle. Best-fit parameter value are represented as points. Parameter values (x-axis) are displayed on log10 scale. Below each profile, the dependencies of all other parameters on the profiled parameter are plotted. Coupling strength is quantified by the relative change of a secondary parameter from its best-fit value after re-optimizing all parameters at each fixed value of the profiled parameter (x-axis). Both axes are shown on a $\log_{10}$ scale. The strongest dependency is highlighted in red, the next strongest in blue, and the remaining dependencies in gray.

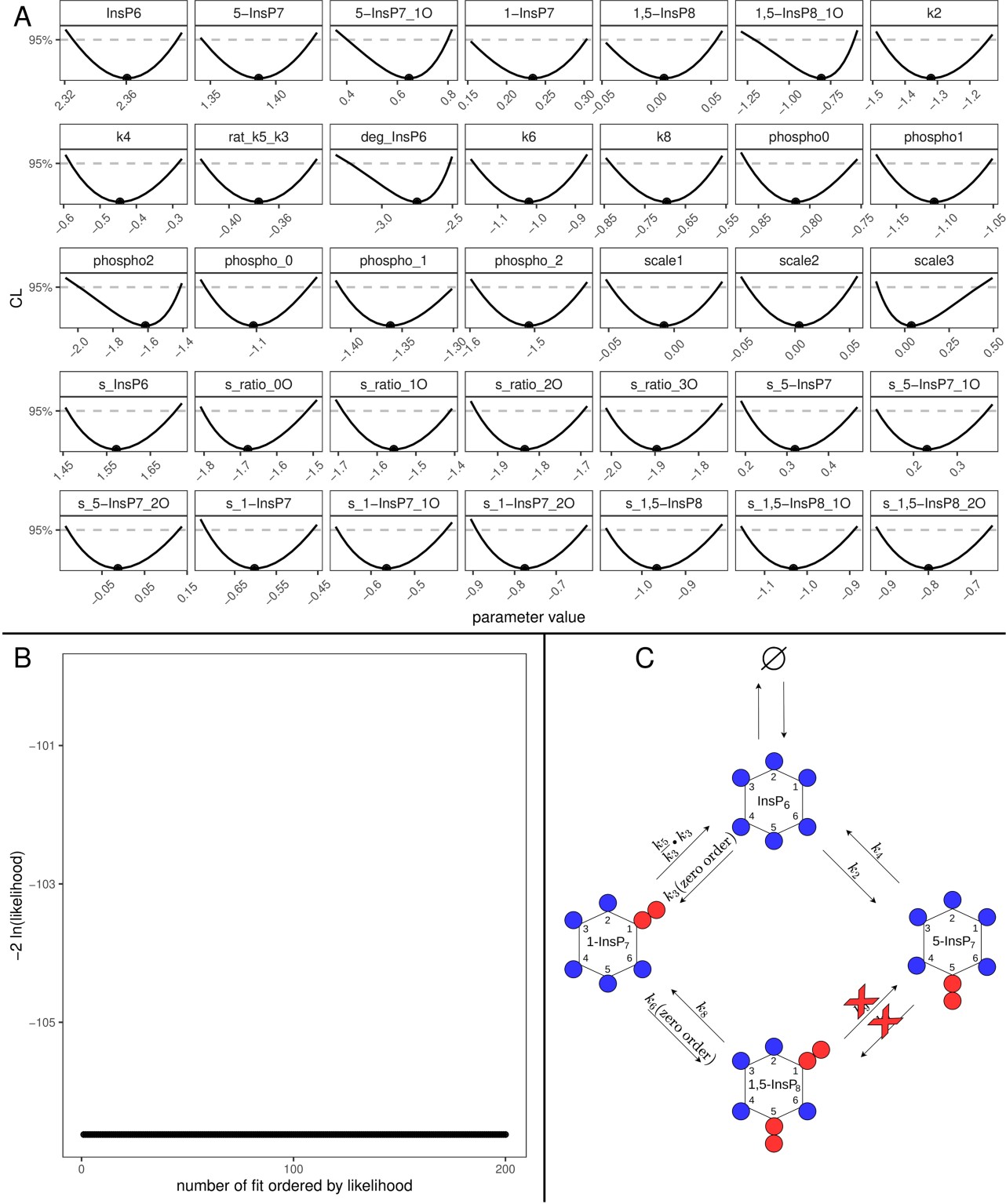

**Fig 11. Fully reduced HCT116 model.** (A) Profiles of the best-fit parameters of the reduced model (line). Best-fit parameter value are represented as points. Parameter values (x-axis) are displayed on log10 scale. (B) Likelihood values of the 200 best multi-start fits, ordered by lowest likelihood value. (C) Representation of the statistically favoured transition scheme, displaying a chain-like pattern rather than a cycle.

For HCT116, a second data set was generated and analysed, where the experimental conditions of the yeast experiment were copied, regarding the transition from low to high phosphate. The fits and the model reduction can be found in the supplement (S1 Fig–S4 Fig). The results are consistent with the analysis of the first HCT116 dataset in the sense that also in this reduced model, the transitions between 5-InsP$_7$ and 1,5-InsP$_8$ are removed. The transition between 1-InsP$_7$ and InsP$_6$ is also removed.

## Discussion

In this study, we investigated the metabolic pathways of inositol pyrophosphates (IPPs) in two different biological systems: yeast and the human tumour cell line HCT116. By utilising pulse-labelling experiments with $^{18}$O water and ordinary differential equation (ODE) models, we explored the synthesis and turnover of IPPs, aiming to clarify the pathways involved in the production of the highly phosphorylated IPP, 1,5-InsP$_8$. Previous studies had proposed that 1,5-InsP$_8$ could be synthesised via two distinct routes: either through 5-InsP$_7$ or 1-InsP$_7$ [16,29,42,55,56]. Our findings challenge this notion, suggesting that the metabolic networks in both yeast and HCT116 cells do not form a cyclic pathway but follow a linear reaction sequence.

We first applied a comprehensive ("full") ODE model to describe the dynamics of IPP metabolism in yeast and HCT116 cells. The initial fitting results revealed that while the model could adequately capture the data, several parameters were not identifiable, indicating redundancies and dependencies that made it challenging to precisely estimate certain reaction rates and Michaelis–Menten constants. To address these issues, we performed a systematic model reduction process, heavily based on the profile-likelihood method. This process involved simplifying the model by reducing certain reactions to mass-action kinetics or even setting specific reaction rates to zero, based on their unidentifiability and lack of contribution to improving the model's fit to the experimental data.

For the yeast system, the model reduction process led to significant insights. We found that several reactions initially assumed to be part of the metabolic cycle could be eliminated without compromising the model's ability to describe the experimental data. Specifically, the pathway from InsP$_6$ to 1,5-InsP$_8$ via 1-InsP$_7$ was not required, and instead, the data were best explained by a linear pathway where 1,5-InsP$_8$ is produced directly from 5-InsP$_7$. This finding suggests that, in yeast, the conversion of InsP$_6$ to 1,5-InsP$_8$ primarily occurs through the phosphorylation of 5-InsP$_7$, with the dephosphorylation of 1,5-InsP$_8$ being the main route for generating 1-InsP$_7$.

This prediction was tested against independent experimental data. Analysis of yeast knockout strains (vip1$\Delta$, kcs1$\Delta$) under the same high phosphate conditions showed that the vip1$\Delta$ mutant accumulated 5-InsP$_7$ and lost 1,5-InsP$_8$, as expected, whereas the kcs1$\Delta$ mutant exhibited no detectable 1-InsP$_7$. The latter observation is inconsistent with the canonical cyclic model, which predicts that Vip1 should provide an alternative route from InsP$_6$, but it is fully consistent with the reduced linear topology. Thus, the knockout data provide biological validation of the break in the cycle predicted by model reduction.

The different reduction outcomes in yeast and human cells also provide mechanistic insight into how enzyme properties shape flux distributions *in vivo*. In yeast, model reduction consistently eliminated the InsP$_6$ → 1-InsP$_7$ branch, leaving 1,5-InsP$_8$ synthesis to proceed through the 5-InsP$_7$ route. This agrees with *in vitro* studies showing that Vip1/PPIP5K has higher catalytic efficiency toward InsP$_7$ compared to InsP$_6$ [42,45,57,58], suggesting that the same preference governs flux allocation *in vivo*. By contrast, in HCT116 cells the 5-InsP$_7$ → 1,5-InsP$_8$ step was eliminated, leaving 1-InsP$_7$ as the principal precursor of 1,5-InsP$_8$. This outcome was unexpected given existing biochemical data [42,45,55,58], and it points to species-specific differences in the relative contributions of IP6Ks and PPIP5Ks. Such differences may reflect divergent regulation of these enzymes or distinct physiological roles of 1- versus 5-InsP$_7$ across eukaryotes.

In the HCT116 cell line, the analysis was more complex due to the presence of two different experimental conditions under which datasets were generated. Both conditions provided valuable insights and led to comparable yet slightly different results. The model reduction process revealed a preference for a linear rather than a cyclic metabolic pathway

under both conditions. However, there were subtle differences in the parameter estimates and the specific reaction rates. While in the "normal to normal" condition, i.e. under constant $P_i$ concentration, only the transitions between 5-InsP$_7$ and 1,5-InsP$_8$ are removed via the model reduction approach, in the "low to high" phosphate condition, the transition from 1-InsP$_7$ to InsP$_6$ is additionally removed. Furthermore in the "normal to normal" condition, the conversion 1-InsP$_7$ to 1,5-InsP$_8$ is reduced to a zero-order reaction, while in the "high to low" setting it is the other way around. This suggests that the linear pathway might be modulated differently depending on the nutrient changes that the cells experience. Despite these differences, in both regimes the cycle is broken at the same point, reinforcing the biological relevance of this topology.

The reliance on Michaelis–Menten kinetics is another point that requires careful interpretation. This rate law cannot capture more complex behaviours such as cooperative binding or allosteric regulation. At the same time, it is a pragmatic and widely used approximation in metabolic modelling, particularly when detailed *in vivo* rate laws are unavailable. In our framework, Michaelis–Menten kinetics functions as a flexible starting point: through model reduction, the data themselves determine whether a reaction effectively reduces to mass-action, zero-order, or remains Michaelis–Menten. This allows the analysis to remain tailored to the information content of the data.

A further limitation is that statistical model reduction may discard low-flux reactions that are biologically important but experimentally difficult to resolve. However, the combination of statistical evidence and knockout validation mitigates this concern: the break predicted by the model in yeast was independently observed in the *kcs1Δ* mutant, which exhibited IPP concentrations incompatible with the cyclic model. Thus, while low-level fluxes below detection thresholds cannot be excluded, the convergence of data-driven reduction and genetic perturbation strengthens confidence in the linear topology as the major route under the studied conditions.

Finally, we took steps to ensure that the reduced models are statistically well defined and not artifacts of local minima. Before reduction, only a minority of runs reached the global optimum, while after reduction the vast majority converged to a single lowest-likelihood solution. Waterfall plots and parameter distribution plots clearly demonstrate this stabilization. In addition, the reduced models exhibit fully identifiable parameter profiles, showing that each parameter is supported by the data. These diagnostics provide strong evidence that the reduced models are robust within our modelling framework.

Interestingly, despite quantitative differences between conditions in HCT116 cells and between yeast and human cells, the consistency in the overall pathway structure underscores the robustness of the linear metabolic model. This finding challenges the traditional view of a cyclic IPP metabolic pathway and suggests that the linear pathway might be a more general feature of IPP metabolism in eukaryotic cells.

Our analysis also highlighted the importance of using both experimental data and mathematical modelling to understand complex biochemical networks. The use of pulse-labelling with $^{18}$O water provided dynamic information on the turnover rates of different IPPs, which was critical for parameter estimation in our ODE models. Additionally, the model reduction approach allowed us to identify the essential reactions in the metabolic pathway, leading to a more parsimonious and interpretable model that still accurately described the experimental data. Notably, in yeast the predictions from the model reduction were directly validated by independent knockout experiments, providing orthogonal support for the inferred topology.

In conclusion, this study provides new insights into the metabolic pathways of IPP conversion, revealing that the synthesis of 1,5-InsP$_8$ in both yeast and HCT116 cells follows a linear rather than a cyclic pathway. The observation that different experimental conditions in HCT116 cells led to comparable, yet slightly distinct, results suggests that while the linear pathway is robust, its regulation may vary under different physiological contexts. These findings have important implications for our understanding of IPP metabolism and its regulation in eukaryotic cells. Further studies will be needed to explore the functional consequences of this linear pathway and to determine whether similar metabolic networks exist in other organisms or under varying physiological conditions.

## Materials and methods

### Mathematical modelling

The modelling approach presented in this work is based on ODEs where every dephosphorylation, or phosphorylation reaction between the considered InsP species, for all of which experimental data has been provided, was modelled according to Michaelis-Menten kinetics

$$\dot{P} = k \cdot \frac{S}{K_m + S} \tag{1}$$

where $P$ is the product concentration, $S$ is the substrate concentration and $K_M$ is the Michaelis-Menten constant. These can be divided into three regimes, namely: **Zero-order reactions**, where the value of $K_M$ is small compared to $S$ and can be neglected, $S$ cancels out and the reaction velocity $\dot{S}$ is independent from the substrate concentration $S$; **Michaelis-Menten reactions**, where $S$ is in the range of $K_M$ and the unreduced Michaelis-Menten kinetic has to be applied; **Mass-action reactions**, where $K_M$ is large compared to $S$ and the contribution of $S$ can thus be neglected in the denominator, meaning the reaction velocity depends linearly on $\frac{k}{K_m}$ and $S$. *In vitro* the considered dephosphorylation, or phosphorylation reactions can be performed by multiple enzymes, but since in this work only the kinetics of the conversions as a whole are of interest, we consider the sum of the possible enzyme contributions and only model *net* reactions by a single Michaelis-Menten reaction. To account for the availability of labelled ATP and to describe the labelling of the different molecules, the phosphorylation reaction velocities $k_i$ are multiplied with the relative abundance of the labelled ATP $ratio_{XO}$ where $X \in \{0, 1, 2, 3\}$ designates the number of labelled oxygen atoms in the $\gamma$-phosphate, for which experimental data is also available. In this analysis we considered two cell types, namely baker's yeast and human HCT116 cells which were both analysed with the identical initial model consisting of 29 states and 48 dynamic parameters. The reactions of the initial full model are shown in Table 1.

An absolute error model was assumed, with error parameters estimated simultaneously with the kinetic parameters. To assess the validity of this assumption, we performed a residuals-versus-prediction analysis for the yeast dataset (S5 Fig). The plot shows no systematic fanning of residuals with increasing predicted values, supporting the use of an absolute error model. Because the same measurement technique was applied in all datasets, this validation was performed once and taken to apply across all remaining datasets. The estimated errors are depicted as grey bands around the plotted predictions for every dataset. The observables and corresponding error parameters are summarized in Table 2. For all parameters, a logarithmic transformation was performed prior to optimization to ensure positivity and to allow efficient sampling across a wide dynamic range.

### Parameter estimation and selection of mathematical models

In order to determine the optimal parameter set, the local deterministic Gauss-Newton gradient-based trust-region optimiser with a tolerance setting of $10^{-10}$ was used, which is implemented in the *R* package *dMod* [47]. To ensure, that the "best" parameter set which is found is also the globally optimal set, a multi-start analyses with 1000 random initial parameters sets was performed. To determine how well a fit describes the analysed data the Bayesian information criterion (BIC) is used in this work. The BIC is defined as $\text{BIC} = k \ln(n) - 2 \ln(\mathcal{L})$, where $k$ is the number of model parameters, $n$ is the number of data points and $\mathcal{L}$ is the maximised value of the likelihood function. Lower BIC values indicate preferable description of the analysed data. Thus, when two models describe the same data comparably well, the model with the lower number of parameters, following Occam's razor, is to be preferred. It should be mentioned that there exists another popular metric of comparing two models called Akaike information criterion (AIC), defined as $\text{AIC} = 2k - 2 \ln(\mathcal{L})$. However, in this work we use BIC since it is a more stringent criterion and AIC under-penalizes the number of parameters if $n$ becomes large. It should be mentioned, that the BIC value is only useful as a relative comparison between different models, while the absolute value of the BIC has no real meaning for any given model.

**Table 1**. **Reactions of the full unreduced mathematical model.** The reactions are based on Michaelis-Menten and mass-action kinetics, where X and X' specify the number of heavy labelled oxygen in the respective phosphate groups, with X and X'$\in \{0, 1, 2, 3\}$.

| Educt | Product | Rate |
|---|---|---|
| $\varnothing$ | $\text{InsP}_6$ | $\text{prod}_{\text{InsP}_6}$ |
| $\text{InsP}_6$ | $\varnothing$ | $\text{deg}_{\text{InsP}_6} \cdot \text{InsP}_6$ |
| $\text{InsP}_6$ | $5\text{-InsP}_{7,XO}$ | $\dfrac{k_2 \cdot \text{ratio}_{XO} \cdot \text{InsP}_6}{Km_2 + \text{InsP}_6}$ |
| $\text{InsP}_6$ | $1\text{-InsP}_{7,XO}$ | $\dfrac{k_3 \cdot \text{ratio}_{XO} \cdot \text{InsP}_6}{Km_3 + \text{InsP}_6}$ |
| $5\text{-InsP}_{7,XO}$ | $\text{InsP}_6$ | $\dfrac{k_4 \cdot 5\text{-InsP}_{7,XO}}{Km_4 + 5\text{-InsP}_{7,XO}}$ |
| $1\text{-InsP}_{7,XO}$ | $\text{InsP}_6$ | $\dfrac{k_5 \cdot 1\text{-InsP}_{7,XO}}{Km_5 + 1\text{-InsP}_{7,XO}}$ |
| $1\text{-InsP}_{7,XO}$ | $1,5\text{-InsP}_{8,XOX'O}$ | $\dfrac{k_6 \cdot 1\text{-InsP}_{7,XO} \cdot \text{ratio}_{X'O}}{Km_6 + 1\text{-InsP}_{7,XO}}$ |
| $5\text{-InsP}_{7,XO}$ | $1,5\text{-InsP}_{8,1X'O+5XO}$ | $\dfrac{k_7 \cdot 5\text{-InsP}_{7,XO} \cdot \text{ratio}_{X'O}}{Km_7 + 5\text{-InsP}_{7,XO}}$ |
| $1,5\text{-InsP}_{8,XOX'O}$ | $1\text{-InsP}_{7,XO}$ | $\dfrac{k_8 \cdot 1,5\text{-InsP}_{8,XOX'O}}{Km_8 + 1,5\text{-InsP}_{8,XOX'O}}$ |
| $1,5\text{-InsP}_{8,XOX'O}$ | $5\text{-InsP}_{7,X'O}$ | $\dfrac{k_9 \cdot 1,5\text{-InsP}_{8,XOX'O}}{Km_9 + 1,5\text{-InsP}_{8,XOX'O}}$ |
| $\text{ratio}_{0O}$ | $\text{ratio}_{1O}$ | $\text{phospho}_0 \cdot \text{ratio}_{0O}$ |
| $\text{ratio}_{0O}$ | $\text{ratio}_{2O}$ | $\text{phospho}_1 \cdot \text{ratio}_{0O}$ |
| $\text{ratio}_{0O}$ | $\text{ratio}_{3O}$ | $\text{phospho}_2 \cdot \text{ratio}_{0O}$ |
| $\text{ratio}_{1O}$ | $\text{ratio}_{0O}$ | $\text{phospho}_{-0} \cdot \text{ratio}_{1O}$ |
| $\text{ratio}_{2O}$ | $\text{ratio}_{0O}$ | $\text{phospho}_{-1} \cdot \text{ratio}_{2O}$ |
| $\text{ratio}_{3O}$ | $\text{ratio}_{0O}$ | $\text{phospho}_{-2} \cdot \text{ratio}_{3O}$ |

**Table 2**. **Observables and error parameters of the full unreduced mathematical model.**

| Dataset | Observable | Error parameter |
|---|---|---|
| $\text{InsP}_6$ | $\text{InsP}_6$ | $s\_\text{InsP}_6$ |
| $5\text{-InsP}_7$ | $5\text{-InsP}_7$ | $s\_5\text{-InsP}_7$ |
| $5\text{-InsP}_{7,1O}$ | $5\text{-InsP}_{7,1O}$ | $s\_5\text{-InsP}_{7,1O}$ |
| $5\text{-InsP}_{7,2O}$ | $5\text{-InsP}_{7,2O}$ | $s\_5\text{-InsP}_{7,2O}$ |
| $1\text{-InsP}_7$ | $1\text{-InsP}_7$ | $s\_1\text{-InsP}_7$ |
| $1\text{-InsP}_{7,1O}$ | $1\text{-InsP}_{7,1O}$ | $s\_1\text{-InsP}_{7,1O}$ |
| $1,5\text{-InsP}_8$ | $1,5\text{-InsP}_8$ | $s\_1,5\text{-InsP}_8$ |
| $1,5\text{-InsP}_{8,1O}$ | $1,5\text{-InsP}_{8,1O0O} + 1,5\text{-InsP}_{8,0O1O}$ | $s\_1,5\text{-InsP}_{8,1O}$ |
| $1,5\text{-InsP}_{8,2O}$ | $1,5\text{-InsP}_{8,2O0O} + 1,5\text{-InsP}_{8,0O2O} + 1,5\text{-InsP}_{8,1O1O}$ | $s\_1,5\text{-InsP}_{8,2O}$ |
| $\text{ratio}_{0O}$ | $\text{ratio}_{0O}$ | $s\_\text{ratio}_{0O}$ |
| $\text{ratio}_{1O}$ | $\text{scale}_1 \cdot \text{ratio}_{1O}$ | $s\_\text{ratio}_{1O}$ |
| $\text{ratio}_{2O}$ | $\text{scale}_2 \cdot \text{ratio}_{2O}$ | $s\_\text{ratio}_{2O}$ |
| $\text{ratio}_{3O}$ | $\text{scale}_3 \cdot \text{ratio}_{3O}$ | $s\_\text{ratio}_{3O}$ |

## Profile likelihood

Analysing parameter uncertainties as well as identifiability of the latter is a key task in mathematical modelling. While there are other methods, like Markov-Chain-Monte-Carlo methods (MCMC), to analyse parameter uncertainty, in this project the profile likelihood method was employed [49,50]. The profile likelihood is a Likelihood ratio test with one degree of freedom, where the analysed parameter is slightly changed from its best-fit value and all remaining parameters are re-optimised. The ratio between the initial best likelihood value and the new one is then plotted. This process is performed multiple times in a set interval for the analysed parameter, usually +/-3 around the best-fit value on log-scale, and ideally will then give a bell shaped curve passing a 95% confidence level threshold for larger and smaller values of the best-fit value of the parameter. This method cannot only be used to investigate whether model parameters are identifiable, but

also provides statistically sound estimates for parameter uncertainties. Furthermore, this method can also guide model reduction by providing not only the parameter uncertainties but also their dependencies on the other parameters in the model. The dependencies depicted under the profile likelihood plots show the relative change of the other parameters from their best-fit value as a function of the relative change of the profiled parameter's value from its best-fit estimate after refitting. To perform the uncertainty analysis in *R*, the profile likelihood function of the *dMod* package was used.

## Yeast strains and genetic manipulation

The Saccharomyces cerevisiae BY4741 was used in this study. The plasmid used to overexpress the truncated Pho90 (Pho90$^{\Delta SPX}$) was generated by cloning the Pho90 open reading frame excluding N-terminal 375 amino acids between BamHI and XhoI restriction enzyme sites of the parent plasmid (pRS415) (Mumberg, Muller, and Funk, 1995) containing TEF1 promoter. For DNA transformation, BY4741 strain was logarithmically grown at 30 °C in 50 mL of YPD (yeast extract-peptone-dextrose) medium. Cultures (4.6 x $10^7$ cells/mL) were centrifuged at 3200 g for 3 min. The pellet was washed with the same volume of TE buffer (10 mM of Tris-HCl pH 7.5 and 1 mM of EDTA) and resuspended with 10 mL of LiAC-TE buffer (TE buffer with 100 mM of LiAC). After the incubation at room temperature for 15 min, cells were collected by centrifugation at 3,000 g for 3 min. The pellet was resuspended with 800 $\mu$L of LiAC-TE buffer. 40 $\mu$L of cells were mixed with 200 ng of plasmid DNA, 5 $\mu$L of salmon sperm DNA (10 mg/mL) and 300 $\mu$L of PEG-LiAC-TE buffer (50% of polyethylene glycol 4000). After 20 min at 30 °C with gentle shaking, a heat shock was carried out at 42 °C for 20 min with gentle shaking. Cells were centrifuged at 3,000 g for 2 min, washed twice with water, and grown on selection plates for 2-4 days at 30 °C. The *kcs1Δ* and *vip1Δ* mutants were obtained from the strains used in [29]. For yeast growth, synthetic complete (SC) medium was prepared from yeast nitrogen base without phosphate (Formedium, UK). The phosphate concentration was adjusted with $KH_2PO_4$. Potassium concentration was controlled by adding KCl instead of $KH_2PO_4$.

## Extraction of IPPs and ATP from yeast cells

Yeast strain overexpressing Pho90$^{\Delta SPX}$ was grown at 20 °C in SC medium containing 0.2 mM of phosphate until logarithmic phase (4.6 x $10^7$ cells/mL). Samples were harvested by centrifugation at 3,000 g for 3 min and resuspended in the same volume of SC medium containing 50 mM of phosphate prepared with 50 % of $^{18}$O-labelled water. Yeast cells were further incubated at 20 °C. At each time point, 3 ml of yeast culture was mixed with 300 $\mu$l of 11 M perchloric acid to a final concentration of 1 M and snap frozen in liquid nitrogen. IPPs and ATP were extracted from yeast samples based on [52]. Thawed samples were centrifuged at 20,000 g for 3 min at 4 °C and the soluble supernatant was transferred into a new tube. 6 mg of titanium dioxide ($TiO_2$) beads per sample (GL Sciences, Japan) was washed twice with 1 mL of water and 1 M of perchloric acid respectively, and then mixed with the soluble supernatant [59]. The mixture was gently rotated for 15 min at 4 °C and centrifuged at 20,000 g for 3 min at 4 °C. After two rounds of washing with 500 $\mu$L of 1 M perchloric acid, the $TiO_2$ beads were incubated with 300 $\mu$L of 3 % (v/v) $NH_4OH$ at 25 °C for 5 min with gently shaking to elute IPPs and ATP. The samples were centrifuged at 20,000 g for 1 min and the eluents were transferred into a new tube. The additional centrifugation was conducted to completely remove the residual $TiO_2$ beads from the eluents. The resulting supernatant was dried in a SpeedVac (Labogene, Denmark) at 42 °C. Samples were kept at -20 °C until analysis.

For mutant analysis, cells were grown in SC medium to logarithimic phase. A total of 5 OD$_{600}$ units of cells were harvested by centrifugation at 3,000 g for 2 min, resuspended in 1 mL of 1 M perchloric acid, and snap-frozen in liquid nitrogen. IPPs were then extracted as described above. Dried samples were reconstituted in 30 $\mu$L of water before analysis.

All time points were generated in three biological replicates.

## Mammalian cell culture and extraction of IPPs and ATP

IPPs and ATP data from HCT116 cells under normal-to-normal phosphate ($P_i$) conditions were obtained from our previously published work [52], while data under low-to-high $P_i$ conditions were generated in the present study. The cell culture preparation and IPP extraction were performed following similar procedures with modifications as described in our previous work. HCT116 cells were cultured for six passages in complete DMEM (Thermo Fisher Scientific, Cat. No. 31966021) supplemented with 10% (v/v) fetal bovine serum (PAN Biotech, Cat. No. P30-3306) and 100 U/mL penicillin-streptomycin (Thermo Fisher Scientific). Cells were grown at 37°C in a humidified incubator with 5% $CO_2$ and 98% humidity. The low phosphate medium (0.05 mM $P_i$) was prepared using $P_i$-free DMEM (Thermo Fisher Scientific, Cat. No. 11971025) supplemented with 100 U/mL penicillin-streptomycin, 1 mM sodium pyruvate, and 0.05 mM phosphate ($NaH_2PO_4$ ). For high phosphate medium containing 50% $^{18}O$ water, a 2x concentrated medium was prepared from DMEM powder (Sigma, Cat. No. D5648) supplemented with 100 U/mL penicillin-streptomycin, 28.16 mM $P_i$ (total 30 mM $P_i$, including 1.84 mM $P_i$ from the DMEM), and 7.4 g/L sodium bicarbonate. This 2x medium was then diluted with an equal volume of $^{18}O$-labelled water to yield a final medium containing 50% $^{18}O$ water. For the low-to-high $P_i$ experiments, HCT116 cells were seeded in complete DMEM and allowed to adhere for 24 hours. Upon reaching approximately 80% confluency, the cells were cultured in low $P_i$ medium (0.05 mM $P_i$) for 24 hours, and then shifted to high $P_i$ medium (15 mM $P_i$) prepared with 50% $^{18}O$ water. At each time point, the medium was removed, and the cells were harvested in 5 mL of 1 M perchloric acid. After snap-freezing in liquid nitrogen, the samples were centrifuged at 3,200 xg for 5 minutes. The soluble supernatant was transferred to a new tube and mixed with $TiO_2$ beads (5 mg per sample). IPPs and ATP were extracted as described previously for yeast cells. All time points were generated in three biological replicates.

## Supporting information

**S1 Fig. Analysis of the HCT116 data-set with the "full" mathematical model model.** (A) Fit of the mathematical model with the best-fit parameters (line) to the experimental data (points). The values are given in $\mu M$ except for the four species in the lower left of the panel which represent ratios of 1,2,3-labelled to 0-unlabelled $\gamma$-phosphate. (B) Likelihood values of the 200 best multi-start fits, ordered by lowest likelihood value. (C) Profiles of the best-fit parameters of the full model (line). Best-fit parameter value are represented as points. Parameter values (x-axis) are displayed on log10 scale. (TIFF)

**S2 Fig. First reduction step in the HCT116 model.** Profiles of the parameters considered for the first model reduction step in the yeast metabolic cycle. Best-fit parameter value are represented as points. Parameter values (x-axis) are displayed on log10 scale. Below each profile, the dependencies of all other parameters on the profiled parameter are plotted. Coupling strength is quantified by the relative change of a secondary parameter from its best-fit value after re-optimizing all parameters at each fixed value of the profiled parameter (x-axis). Both axes are shown on a $\log_{10}$ scale. The strongest dependency is highlighted in red, the next strongest in blue, and the remaining dependencies in gray. (TIFF)

**S3 Fig. Second and third reduction steps in the HCT116 model.** Profiles of the parameters considered for the second (A) and third (B) model reduction step in the yeast metabolic cycle. Best-fit parameter value are represented as points. Parameter values (x-axis) are displayed on log10 scale.Below each profile, the dependencies of all other parameters on the profiled parameter are plotted. Coupling strength is quantified by the relative change of a secondary parameter from its best-fit value after re-optimizing all parameters at each fixed value of the profiled parameter (x-axis). Both axes are shown on a $\log_{10}$ scale. The strongest dependency is highlighted in red, the next strongest in blue, and the remaining dependencies in gray. (TIFF)

**S4 Fig. Fully reduced HCT116 model.** (A) Profiles of the best-fit parameters of the reduced model (line). Best-fit parameter value are represented as points. Parameter values (x-axis) are displayed on log10 scale. (B) Likelihood values of the 200 best multi-start fits, ordered by lowest likelihood value. (C) Representation of the statistically favoured transition scheme, displaying a chain-like pattern rather than a cycle, with an additional removed transition between 1-InsP$_7$ and InsP$_6$ compared to the normal-to-normal HCT116 data set.
(TIFF)

**S5 Fig. Absolute residuals (value of datapoint minus value of its prediction) plotted against predicted values for each observable.** Dashed horizontal lines indicate zero residuals. A roughly constant spread of residuals across the range of predictions supports the assumption of homoscedasticity, and thus the use of an absolute error model.
(TIFF)

**S6 Fig. Absolute residuals (value of datapoint minus value of its prediction) plotted against time-points for each observable.** Dashed horizontal lines indicate zero residuals. A roughly constant spread of residuals across the range of time-points shows that fit is not unduly driven by certain noisy time-points.
(TIFF)

**S7 Fig. Parameter distributions of the 200 best fits in the lowest step of the reduced yeast model.** Parameter values are displayed on log10 scale and follow a point-like distribution with no discernable variance over 200 fits.
(TIFF)

**S8 Fig. Parameter distributions of the 200 best fits in the lowest step of the reduced HCT116 normal to normal P$_i$ model.** Parameter values are displayed on log10 scale and follow a point-like distribution with no discernable variance over 200 fits.
(TIFF)

**S9 Fig. Parameter distributions of the 200 best fits in the lowest step of the reduced HCT116 low to high P$_i$ model.** Parameter values are displayed on log10 scale and follow a point-like distribution with no discernable variance over 200 fits.
(TIFF)

## Author contributions

**Conceptualization:** Andreas Mayer, Henning Jessen, Jens Timmer.

**Data curation:** Geun-Don Kim, Guizhen Liu.

**Formal analysis:** Jacques Hermes.

**Funding acquisition:** Andreas Mayer, Henning Jessen, Jens Timmer.

**Investigation:** Jacques Hermes, Geun-Don Kim, Guizhen Liu.

**Methodology:** Jacques Hermes.

**Project administration:** Jacques Hermes, Andreas Mayer, Henning Jessen, Jens Timmer.

**Resources:** Geun-Don Kim, Guizhen Liu, Maria Giovanna De Leo, Andreas Mayer, Henning Jessen.

**Software:** Jacques Hermes.

**Supervision:** Andreas Mayer, Henning Jessen, Jens Timmer.

**Validation:** Geun-Don Kim, Guizhen Liu.

**Visualization:** Jacques Hermes.

**Writing – original draft:** Jacques Hermes, Geun-Don Kim, Andreas Mayer, Henning Jessen, Jens Timmer.

**Writing – review & editing:** Jacques Hermes, Andreas Mayer, Henning Jessen, Jens Timmer.

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
