## [Decision Letter · Decision Letter 0]

24 Jul 2025

PCOMPBIOL-D-25-01024

The break in the cycle: Inositol pyrophosphate fluxomics disentangled via mathematical modelling

PLOS Computational Biology

Dear Dr. Hermes,

Thank you for submitting your manuscript to PLOS Computational Biology. After careful consideration, we feel that it has merit but does not fully meet PLOS Computational Biology's publication criteria as it currently stands. Therefore, we invite you to submit a revised version of the manuscript that addresses the points raised during the review process.

Please submit your revised manuscript within 60 days Sep 23 2025 11:59PM. If you will need more time than this to complete your revisions, please reply to this message or contact the journal office at ploscompbiol@plos.org. Please include the following items when submitting your revised manuscript:

We look forward to receiving your revised manuscript.

Kind regards,

Marc Birtwistle

Section Editor

PLOS Computational Biology

**Additional Editor Comments :**

Both reviewers raised substantial concerns regarding biological insight and interpretability. The revision should be substantial to make sure these critical points for a publishable manuscript are addressed.

**Journal Requirements:**

At this stage, the following Authors/Authors require contributions: Jacques Hermes, Timmer Jens, Kim Geun-Don, De Leo Maria Giovanna, Mayer Andreas, Liu Guizhen, and Jessen Henning. Please ensure that the full contributions of each author are acknowledged in the "Add/Edit/Remove Authors" section of our submission form.

4) We noted that you indicated that "The fits and the model reduction can be found in the supplement (Figs.10-13). If the figures are supplemental, please remove them from the manuscript and upload them with the file type 'Supporting Information'.

Please ensure that each Supporting Information file has a legend listed in the manuscript after the references list. Please also cite and label them as as "S1 Figure", S2 Figure" and so forth. 

1) State the initials of the authors who received these grants "499552394 and INST 35/159".

3) If any authors received a salary from any of your funders, please state which authors and which funders.

6) Please ensure that the funders and grant numbers match between the Financial Disclosure field and the Funding Information tab in your submission form. Note that the funders must be provided in the same order in both places as well. Currently, "the Deutsche Forschungsgemeinschaft (DFG) under Germany's excellence strategy (CIBSS, EXC-2189, Project ID 390939984" is missing from the Funding Information tab.

7) Thank you for stating "The authors declare that they have no conflict of interest." Please modify your Competing Interest statement to the standard "The authors have declared that no competing interests exist" at resubmission.

**Reviewers' comments:**

Reviewer's Responses to Questions

Reviewer #1: This manuscript presents a novel study by challenging the cyclic model of IPP metabolism and proposing a linear pathway using a sophisticated combination of pulse-labeling and ODE-based modeling. The cross-species analysis and statistical approach (profile likelihood, model reduction) add significant value to the field.

However, the study’s reliance on specific experimental conditions, limited experimental validation, and lack of mechanistic or functional insights are notable shortcomings.

In this regard, the authors should address the following issues:

1.- Validation of linear pathway: there is heavy reliance on mathematical modeling without experimental validation (e.g., genetic knockouts) to confirm the linear pathway.

This should be more explicitely recognized, suggesting future work to validate the findings. For example, which targeted experiments could be carried out to validate the model’s predictions biologically?

2.- Ruling out condition-specific findings: the results seem to be based on specific phosphate conditions and may not generalize to other physiological states. Authors should discuss this possibility and how to test it.

3.- Limited mechanistic insights: authors should provide better explanation of why certain reactions dominate (e.g., enzymatic properties of IP6Ks/PPIP5Ks).

4.- Model reduction: the model reduction process is complex and inaccessible to non-specialists. It would be helpful to add a simplified explanation or visual guide for model reduction steps.

5.- Model assumptions: the study assumes Michaelis-Menten kinetics, potentially oversimplifying IPP metabolism. Authors should discuss the limitations of these assumptions and the role that alternative kinetics could play (e.g., allosteric regulation).

6.- Statistical versus biological relevance: model reduction may overlook low-flux but biologically relevant pathways. How could this scenario be taken into account?

Reviewer #2: Summary:

This manuscript presents a quantitative systems biology framework for analyzing inositol pyrophosphate (IPP) metabolism in yeast and human tumor cell lines. The authors utilize *in vivo* pulse-labeling with H2-18O water and construct ordinary differential equation (ODE) models incorporating Michaelis-Menten kinetics to simulate metabolic fluxes of various IPPs, including 1,5-InsP8, 1-InsP7, and 5-InsP7. Through a multi-start profile likelihood method and successive model reduction, they challenge the canonical view of cyclic flux through IPPs and instead propose a simplified, linear reaction topology for both organisms. While the mathematical rigor and computational reproducibility are commendable, the manuscript suffers from a lack of clarity regarding experimental controls, insufficient robustness diagnostics for global convergence claims, and potentially overzealous structural reduction based on noise-prone data. However, the study holds potential but requires substantial revisions for enhanced biological interpretability, experimental transparency, and mathematical accountability.

Major Critiques:

1. Title and Framing: The phrase "The break in the cycle" is metaphorically appealing but scientifically imprecise. It lacks direct reference to the biological system or the specific reaction cycle under investigation. A more informative and accurate title reflecting the metabolic recharacterization would serve the scientific audience better.

2. Introduction, Lines 35–37: The authors mention that IPPs can induce "non-enzymatic pyrophosphorylation" of proteins such as CK2, AP3, and dynein. However, this concept is not linked to any model assumption or rate law within the paper. If non-enzymatic events are critical to IPP signaling, they should be explicitly addressed or justified as excluded in the model structure.

3. Introduction, Lines 73–87: The rationale for testing the cycle hypothesis is not clearly defined. The authors mention competing routes to 1,5-InsP8 but do not articulate a falsifiable hypothesis. A clearer biological objective would strengthen the modeling aim.

4. Results, Lines 100–113: While the manuscript employs pulse-labeling data for flux inference, there is no description of how many biological or technical replicates were used per time point, though data suggests 3 samples, (biological or technical replicates?). Given the known variability in mass spectrometry-based isotopologue quantification, this is a serious omission. The possibility that noisy signal from low-abundance species (e.g., 1-InsP7, 5-InsP7) influenced model reduction should be explicitly acknowledged.

5. Results, Lines 115–188 (Yeast model): Technically, the dynamic profile fits could be biased to the fitting model which miss out on some experimental data at some timepoints and as such reduce several reactions to zero (e.g., k3, k5) or remove initial values (e.g., singly labeled 5-InsP7). However, this may reflect a lack of experimental signal rather than true biological inactivity. Another example, early time points for 1-InsP7 may carry high coefficient of variation, which flattens profile likelihoods and artificially promotes parameter elimination. No residual plots or CV-aware model fitting diagnostics are provided.

6. Results, Lines 189–226 (HCT116 model): The model reduction pathway differs under two phosphate conditions, yet this is not discussed as a limitation or potential modeling artifact. The claim that linear topology is a universal conclusion is overstated given these divergent results.

7. Figures 2B, 5B, 6B, 9B (Fitness Landscape): The authors present waterfall plots of the top 50 multi-start fits to claim global optimality. However, with ~8–12 free parameters and only 200 multi-starts, this does not satisfy the law of large numbers. A global optimum in nonlinear space typically requires >10,000 samples or complementary diagnostics (e.g., likelihood landscapes, posterior sampling). The number of convergence basins is not quantified.

8. Materials and Methods, Lines 297–312: The error model is described as absolute but is not validated against experimental data. Given variable intensity ranges among isotopologues, a heteroscedastic error model may be more appropriate. Residual distributions or standard deviation bands should be shown to justify the error structure.

9. Mathematical Modeling, Lines 301–309: The transitions from Michaelis-Menten to mass-action or zero-order are mathematically correct but biologically nontrivial. The assumption that KM >> S is not supported by data on actual intracellular concentrations. The authors should provide substrate concentration ranges or simulated bounds.

10. Optimization Protocol, Lines 325–338: While multi-start optimization and log-space transformation are best practices, no convergence criteria are reported (e.g., gradient norm, tolerance thresholds). No variability statistics are shown across 200 runs, and there's no visual check of multimodality or parameter identifiability clusters.

11. Overall: The structural model may be underdetermined by the available data. Excessive parameter reduction based on profile flatness could eliminate biologically relevant fluxes. The conclusions about pathway linearity may be an artifact of model simplification and insufficient experimental resolution.

Minor Critiques:

1. Figures 3 and 7, Lines within Results section: While profile dependencies are well-illustrated, the red/blue/gray color scheme for parameter dependencies needs clearer legends. Parameter coupling should be quantified, not just visualized.

2. Discussion, Lines 233–295: The authors do not critically reflect on the consequences of excluding cyclic routes. Is this behavior conserved across eukaryotes? Could these pathways be condition-specific rather than universally dispensable?

3. Materials and Methods, Line 311: Typo: "netto reactions" should be corrected to "net reactions."

Issue of Grammar, Spelling, Proofreading, etc.

1. General: The writing is clear and mostly free of grammatical issues. However, several terms ("disentangled", "zero order", "profile likelihood") are used without definition. The manuscript would benefit from a glossary or expanded introduction to modeling terminology. There are also few other spots that need clarity… (e.g. line 143, Expect or except?) A thorough edit is required to ensure clarity and readability.

**Have the authors made all data and (if applicable) computational code underlying the findings in their manuscript fully available?**

Reviewer #1: Yes

Reviewer #2: None

PLOS authors have the option to publish the peer review history of their article (what does this mean?). If published, this will include your full peer review and any attached files.

Reviewer #1: No

Reviewer #2: No

**Figure resubmission:**
---

## [Decision Letter · Decision Letter 1]

29 Oct 2025

Dear Mr. Hermes,

We are pleased to inform you that your manuscript 'A Linear Pathway for Inositol Pyrophosphate Metabolism Revealed by ^18^O Labeling and Model Reduction.' has been provisionally accepted for publication in PLOS Computational Biology.

Best regards,

Marc Birtwistle

Section Editor

PLOS Computational Biology

Reviewer's Responses to Questions

**Comments to the Authors:**

Reviewer #1: I find the new revised version satisfactory.

**Have the authors made all data and (if applicable) computational code underlying the findings in their manuscript fully available?**

Reviewer #1: Yes

PLOS authors have the option to publish the peer review history of their article (what does this mean?). If published, this will include your full peer review and any attached files.

Reviewer #1: No

---

## [Editor Report · Acceptance letter]

PCOMPBIOL-D-25-01024R1

A Linear Pathway for Inositol Pyrophosphate Metabolism Revealed by ^18^O Labeling and Model Reduction.

Dear Dr Hermes,

I am pleased to inform you that your manuscript has been formally accepted for publication in PLOS Computational Biology. Your manuscript is now with our production department and you will be notified of the publication date in due course.

With kind regards,

Zsofia Freund
